# Genome-wide enhancer-gene regulatory maps link causal variants to target genes underlying human cancer risk

Pingting Ying[1,2,3,12], Can Chen[1,2,3,12], Zequn Lu[1,2,3,12], Shuoni Chen[1], Ming Zhang[1], Yimin Cai [1], Fuwei Zhang[1], Jinyu Huang[1], Linyun Fan[1], Caibo Ning[1], Yanmin Li[1], Wenzhuo Wang[1], Hui Geng[1], Yizhuo Liu[1], Wen Tian[1], Zhiyong Yang[4], Jiuyang Liu[5], Chaoqun Huang[5], Xiaojun Yang[5], Bin Xu[6], Heng Li[7], Xu Zhu[8], Ni Li [9], Bin Li[1], Yongchang Wei[10], Ying Zhu[1], Jianbo Tian [1,2,3] ✉ & Xiaoping Miao [1,2,3,11] ✉

Genome-wide association studies have identified numerous variants associated with human complex traits, most of which reside in the non-coding regions, but biological mechanisms remain unclear. However, assigning function to the non-coding elements is still challenging. Here we apply Activity-by-Contact (ABC) model to evaluate enhancer-gene regulation effect by integrating multi-omics data and identified 544,849 connections across 20 cancer types. ABC model outperforms previous approaches in linking regulatory variants to target genes. Furthermore, we identify over 30,000 enhancer-gene connections in colorectal cancer (CRC) tissues. By integrating large-scale population cohorts (23,813 cases and 29,973 controls) and multipronged functional assays, we demonstrate an ABC regulatory variant rs4810856 associated with CRC risk (Odds Ratio = 1.11, 95%CI = 1.05–1.16, $P = 4.02 \times 10^{-5}$) by acting as an allele-specific enhancer to distally facilitate *PREX1*, *CSE1L* and *STAU1* expression, which synergistically activate p-AKT signaling. Our study provides comprehensive regulation maps and illuminates a single variant regulating multiple genes, providing insights into cancer etiology.

Genome-wide association studies (GWAS) have identified thousands of loci associated with human traits and diseases[1–3]. Approximately 90% of the variants identified by GWAS fall in non-coding regions such as enhancers and promoters[4]. Fueled by the advanced biochemical investigation of human genome, we are aware that such non-coding variants could affect expression of key genes through regulatory mechanisms, thereby contributing to cancer susceptibility[5]. However, assigning function to the non-coding elements is notoriously difficult,

[1]Department of Epidemiology and Biostatistics, School of Public Health, Wuhan University, Wuhan 430071, China. [2]Department of Gastrointestinal Oncology, Zhongnan Hospital of Wuhan University, Wuhan 430071, China. [3]Department of Radiation Oncology, Renmin Hospital of Wuhan University, Wuhan 430071, China. [4]Department of Hepatobiliary and Pancreatic Surgery, Zhongnan Hospital of Wuhan University, Wuhan 430071, China. [5]Department of Gastrointestinal Surgery, Zhongnan Hospital of Wuhan University, Wuhan University, Wuhan 430071, China. [6]Cancer Center, Renmin Hospital of Wuhan University, Wuhan University, Wuhan 430060, China. [7]Department of Urology, Tongji Hospital of Tongji Medical College, Huazhong University of Science and Technology, Wuhan 430030, China. [8]Department of Gastrointestinal Surgery, Renmin Hospital of Wuhan University, Wuhan 430071, China. [9]Office of Cancer Screening, National Cancer Center/National Clinical Research Center for Cancer/Cancer Hospital, Chinese Academy of Medical Sciences and Peking Union Medical College, Beijing 100021, China. [10]Department of Gastrointestinal Oncology, Hubei Cancer Clinical Study Center, Zhongnan Hospital of Wuhan University, Wuhan 430071, China. [11]Department of Epidemiology and Biostatistics, School of Public Health, Tongji Medical College, Huazhong University of Sciences and Technology, Wuhan 430030, China. [12]These authors contributed equally: Pingting Ying, Can Chen, Zequn Lu. ✉e-mail: tianjb@whu.edu.cn; xpmiao@whu.edu.cn

which needs to distinguish the truly functional enhancers among many transcriptional regulatory sequences and determine the target genes that they variably affect.

Although recent advances in fine-mapping techniques have improved our ability to nominate causal variants, assigning the underlying target genes remains a critical challenge[6,7]. One default approach has been to assign variants to the closest gene at each locus, while predictions based solely on physical proximity alone can be misleading as causal variants are predominantly regulatory variants that can influence gene expression over long genomic ranges[8]. Large gene expression quantitative trait loci (eQTL) data sets have been instrumental for GWAS target genes identification, which add the evidence that majority of the causal genes at GWAS loci are not the closest[8]. However, previously published eQTL datasets could explain only a small fraction (9–13%) of GWAS heritability of cancers[9], suggesting increasingly diverse functional genomic data beyond gene transcription are required to interpret disease mechanisms.

To date, there has been tremendous effort to dissect the machinery of gene regulation including the map of cis-regulatory elements (CREs) from ENCODE project and genome-scale chromosome conformation capture (Hi-C) technology[10–13]. Traditionally, predicting enhancers based on histone ChIP-seq data with H3K27ac enrichment and estimating 3D genome interactions according to Hi-C peaks have been widely utilized respectively. Therefore, integration of multi-omics datasets across a wide range of cell types and tissues is expected to strengthen the signal for pinpointing causal variants and their probable target gene, such as epigenomics data, chromatin accessibility and chromatin interaction datasets. More importantly, compared with the overlapping results of these multi-omics datasets, a computational approach Activity-by-Contact (ABC) model that is developed to link regulatory elements to their target genes with a quantitative combination of enhancer activity and 3D contact frequencies helps to distinguish significant enhancer-gene pairs, further identifying the causal non-coding GWAS variants residing in ABC enhancers and their potential target genes[14,15]. Researchers have created genome-wide maps of more than six million ABC enhancer-gene connections among 74 distinct primary cell types or tissues[15]. However, the systematic landscape for regulatory maps linking enhancers to their potential target genes and those regulatory mechanisms in human cancers have not been fully elucidated.

In this work, we systematically build genome-wide enhancer-gene maps across 20 human cancer types by integrating the multi-omics data using the ABC model. We totally identify 544,849 enhancer-gene connections involving 266,956 enhancers and 216,268 target genes, providing a comprehensive resource of regulatory maps in human cancers. We further characterize functional features both of variants harboring in enhancers and target genes, increasing the understanding for the potential regulatory mechanism of ABC enhancers. It is worth mentioning that ABC model performs better at prediction of regulatory elements and target genes compared with previous known methods. Additionally, we also apply the ABC framework and identify over 30,000 enhancer-gene connections from our CRC tissues. Furthermore, we systematically screen ABC regulatory variants associated with CRC risk in 17,789 cases and 19,951 control and independently validate in a large-scale population consisting of 6024 cases and 10,022 controls. We identify an ABC regulatory variant rs4810856 that acts as an allele-specific enhancer to distally provoke the expression of *PREX1*, *CSE1L* and *STAU1* by ZEB1 mediating long-range chromatin interaction loops, which synergistically activate p-AKT signaling and facilitate CRC cells proliferation in vitro and in vivo, thus contributing to an increased risk of CRC (Odds Ratio (OR) = 1.11, 95%CI = 1.05–1.16, $P = 4.02 \times 10^{-5}$). In summary, we have provided comprehensive insights into genome-wide enhancer-gene connections, linking risk variants to disease genes across multiple cancer types and elucidate a functional model in which a non-coding variant could facilitate long-range chromatin interactions to regulate expression of multiple genes. These findings present a promising approach for distinguishing the causal variants among numerous candidates and predicting target genes, and provide valuable clues for holistic comprehension of the genetic architecture of cancers.

## Results

### Landscapes of genome-wide enhancer-gene maps across 20 cancer types

To map the connections between enhancer and target genes at a genome-wide scale across different cancer types. We predicted the enhancer-gene maps across 20 cancer cell types using ABC model, which combines the enhancer accessibility (ATAC-seq or DNase-seq) and activity (H3K27ac ChIP-seq) with the normalized contact frequency (Hi-C) (Fig. 1a). The data sources for each cancer type were listed in Supplementary Data 1. We totally identified 544,849 enhancer-gene connections for 266,956 enhancers and 216,268 potential effector genes (Fig. 1b–d). The number of identified enhancer-gene connections averaged 27,243, ranging from 20,134 in Uterine corpus endometrial carcinoma (UCEC) to 37,053 in acute myeloid leukemia (LAML). On average, each ABC enhancer was predicted to regulate 2.0 genes, each gene was predicted to be regulated by 2.5 ABC enhancers, and the median genomic distances between each enhancer-gene connection was 28,266 bp (Fig. 1e–g). Notably, we found only 0.5% of enhancer-gene connections were shared among pairs of cancer types, indicating that most regulatory landscapes were highly cancer-type-specific (Fig. 1h), which is consistent with the previous study[16]. To evaluate the prediction ability of ABC enhancers, we quantified the enrichment of variants that functionally validated from a high-throughput reporter assay among multiple cancer types in ABC enhancers[17]. We found that these functionally validated variants were more enriched in ABC enhancers compared with other regulatory elements, such as ATAC peaks, H3K27ac peaks, FANTOM5 enhancers, HiC signal and eQTLs (Fig. 1i). Overall, we demonstrated a promising approach based on multi-omics data to build regulatory maps in multiple cancers.

### Functional characterization of cancer enhancer-gene connections

Accumulating evidence has supported that gene expressions are typically regulated by transcription factors binding genetic variants among enhancers[8,18]. To characterize features of variants resided in the ABC enhancers (ABC variants), we generated the control variants set (non-ABC variants) using a web tool vSampler with the allele frequencies, number of variants in LD, as well as genomic distribution matched to ABC variants for each cancer type. Compared with non-ABC variants, ABC variants were significantly enriched within non-coding regions in human genome, such as TF binding region, 5′UTR and gene upstream region (Fig. 2a). To further demonstrate the potential regulatory function of ABC variants, we analyzed whether these variants were enriched within genomic regions marked by histone modification and TF binding sites. The ChIP-seq data of histone modification and TFs across various cancer cells were downloaded from ENCODE and TFs with at least 50,000 peaks in the ChIP-seq data were finally selected. As expected, we observed significant enrichments of ABC variants within active chromatin regions, including H3K27ac, H3K4me2, H3K9ac, H3K4me1, H3K4me3, H3K36me3 and TF binding sites, rather than repressive epigenetic regions including H3K27me3 (Fig. 2b). Intriguingly, abundant cancer-associated TFs were identified preferentially binding to ABC variants (Fig. 2c), for instance, TEAD4 have been extensively reported that plays important roles in gene expression regulation among cancers[19]. Meanwhile, we analyzed the enrichment of ABC variants among the chromatin state, and found ABC variants were enriched in active chromatin state, including enhancers and active TSS (Fig. S1). Collectively, these

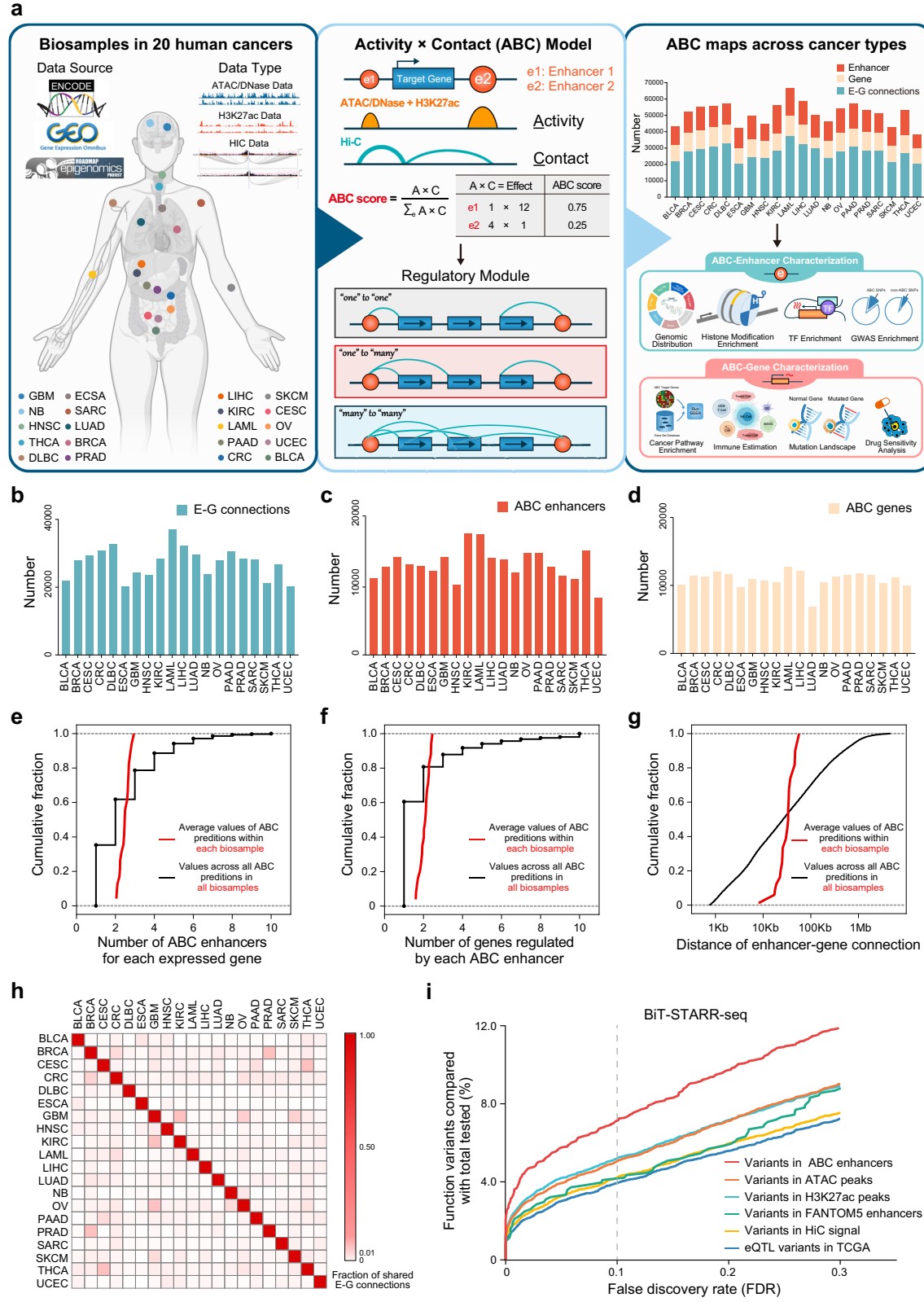

findings provide strong evidence supporting the regulatory properties of ABC variants, which are mediated by TFs.

We further assessed whether ABC variants were enriched for susceptible variants associated with cancers. As a result, ABC variants significantly enriched in GWAS loci among most cancer types, particularly in esophageal cancer (ESCA, $P = 1.43 \times 10^{-13}$), indicating that ABC variants could provide additional insight into cancer heritability (Fig. 2d). Therefore, we performed in-depth analysis using

LD score regression (LDSC) to assess the proportion of ABC variants associated with heritability for each cancer type, and observed that ABC variants could explain a significant fraction of cancer heritability, ranging from 0.5% in thyroid carcinoma (THCA) to 12% in prostate adenocarcinoma (PRAD, Fig. 2e). Additionally, ABC variants presented more significant population-associated $P$ values compared to genome-wide variants in the cancer types, such as CRC and PRAD (Figs. 2f and S2). Taken together, our findings suggest that

**Fig. 1 | Landscapes of genome-wide enhancer-gene maps across 20 cancer types. a** Overview of ABC enhancer-gene maps across 20 cancer types, which was created with BioRender.com. The multi-omics data that were used to build ABC model including DNase-seq, ATAC-seq, H3K27ac ChIP-seq and HiC-seq of 20 cancer types from multiple datasets were indicated at the left. Enhancer-gene connections were identified by the calculation of ABC score for investigating the different gene regulatory modules. ABC variants and target genes were characterized based on functional enrichment compared to non-ABC variants or genes. Summary of the enhancer-gene maps across 20 cancer types. Bar charts represented the number of enhancer-gene connections (**b**), ABC enhancers (**c**) and ABC genes (**d**) in each cancer type. **e** Cumulative fractions of the number of enhancers predicted to regulate each gene across 20 cancer types (black line; mean = 2.5) and the mean number of enhancers predicted to regulate each gene within each cancer type (red line; median = 2.5). **f** Cumulative fractions of the number of genes regulated by each ABC enhancer across 20 cancer types (black line; mean = 2.0) and the mean number of genes regulated by each ABC enhancer within each cancer type (red line; median = 2.1). **g** Cumulative fractions of the genomic distances between the enhancer and the gene for each predicted enhancer–gene connection across 20 cancer types (black line; median = 26,755 bp) and the median genomic distance between each enhancer-gene connection within each cancer type (red line; median = 28,266 bp). **h** Replicability of enhancer-gene pairs across cancer types. The color represents the replication ratio of enhancer-gene pairs of one cancer (y-axis) in another cancer type (x-axis). Two connections are considered overlapping if the predicted genes were the same and the enhancer elements overlapped. **i** Enrichment analyses for variants detected from different approaches (ABC model, ATAC peaks, H3K27ac peaks, FANTOM enhancers, HiC signals or eQTL) within the functionally validated genetic variants set tested by Biallelic Targeted STARR-seq (BiT-STARR-seq) from multiple cancers. The percentages of variants deemed to have genotype-dependent enhancer activity at different FDR thresholds were shown for each set. Source data are provided with this paper.

ABC variants might make a significant contribution to the heritability of multiple cancers.

To further dissert the potential function of the genes regulated by ABC enhancers (ABC genes), we characterized functional roles of the target genes in terms of gene pathway enrichment, tumor mutation burden, drug response and immune infiltration. GSEA analysis revealed that ABC genes were enriched for TNFA signaling pathway via NFKB among most of cancer types (Fig. 2g), which is reported play a critical role in cancer development[20]. Moreover, emerging evidence has demonstrated that genes within somatic mutation and copy-number alterations (SCNAs) burden can act as key oncogenic drivers and serve as an effective approach for target therapy[21,22]. We observed significantly enrichment of ABC genes in somatically mutated genes set compared with non-ABC genes, and found frequently amplification and deletion alterations among ABC genes (Fig. 2h, i). These high frequencies of alterations indicated genes tend to contain regions with driver events associated with cancer development. Interesting, we found that drugs targeting ERK/MAPK signaling were broadly associated with more ABC genes in the drug response analyses based on Genomics of Drug Sensitivity in Cancer database (GDSC) (Fig. S3). Given that growing evidence demonstrates that immune response could greatly influence cancer development[23], we next assessed the association between ABC genes expression and immune cell infiltration. Unsurprisingly, ABC genes were closely related to high infiltration by immune cells among most of cancer types, especially in PRAD. (Figs. 2j and S4a). Furthermore, these genes significantly enriched in immune-related pathways as well, such as EGFR1 pathway, which was much important for cell growth and differentiate (Fig. S4b). Collectively, these integrated analyses indicate that ABC genes could play an important role in tumorigenesis, and act as effective biomarkers or promising therapeutic targets for cancers.

## The ABC model performs better at linking enhancer variants to genes in our CRC tissues

A number of studies have illustrated that variations located enhancer regions can facilitate enhancer-promoter interactions, thus resulting in an increased risk of CRC[8,24,25], To created enhancer-gene maps to determine causal variant of CRC risk, we performed multi-omics analyses of ATAC-seq, H3K27ac ChIP-seq and RNA-seq with high quality from our 10 CRC tissues (Fig. 3a and Fig. S5) and the clinical characteristics of the 10 CRC patients were provided in Supplementary Table 1. By computationally integrating these multi-omics data, we identified 34,130 enhancer-gene connections involving 15,121 unique enhancers and 12,351 expressed genes. On average, each ABC enhancer was predicted to regulate 2.8 genes, each gene was predicted to be regulated by 2.3 ABC enhancers, and the median genomic distances between each enhancer-gene connection was 48,375 bp (Fig. 3b–d). Additionally, the identified ABC enhancers contained 26,877 non-coding variants with underlying regulatory effect (Fig. 3a).

To evaluate the performance of ABC predictions for assigning target genes to regulatory variants. We tested a credible set consisting of 27 variant-gene connections which were validated by functional experiments from previous CRC studies (Supplementary Table 2). Notably, ABC model that identified 16 variant-gene connections out of 19 cases (70% recall and 84% precision), had higher precision than other approaches, including predictions based on genomic position, eQTLs, three-dimensional contacts or other enhancer-gene predictions (Fig. 3e). This analysis demonstrates that ABC regulatory maps could accurately connect fine-mapped variants to target genes, highlighting the performance for regulatory function annotations by ABC model.

We further annotated the position distribution of ABC variants in genome, and observed that about 90% of variants lied in non-coding regions, such as intron and intergenic regions (Fig. 3f). Moreover, we found ABC enhancers exerted higher chromatin activity, and were closer to the closest TSS compared with other accessible regions (Fig. 3g). In parallel with the results of CRC cell line, ABC variants recognized in our tumor tissues were enriched among TF binding sites and active epigenetic makers such as H3K27ac, H3K4me1, ATAC peaks and DNaseI hypersensitive sites (DHSs) (all of $P < 0.05$, Fig. 3h and Fig. S6). These observations suggest that the central role for ABC variants among the transcriptional regulation in CRC development.

To elaborate the potential contribution of ABC genes to CRC development and clinical relevance, we detected 5377 significant differently expressed genes in colorectal cancer tissues and paired normal tissues (Fig. S7a). Subsequently, a series of functional analyses were conducted to characterized these genes. The results showed that ABC genes were closely relevant to cancer signaling pathways, high mutation burden and pharmaceutical targets, which will be helpful for clinical application, which were consisted with the results analyzed in cell lines (Fig. S7b–f). Together, these findings highlight the strength of ABC model in predicting enhancer-gene connections which play critical roles in aberrant regulation of CRC tumorigenesis.

## ABC variants can explain a significant proportion of CRC heritability

To determine the potential role of ABC variants identified from our CRC samples contributing to CRC risk, we performed enrichment analyses using the published GWAS data. Intriguingly, we found that 69 of the 149 CRC GWAS loci were identified by ABC framework and linked to 111genes. It is noteworthy that ABC model enables the identification of putative risk genes outside previously reported CRC loci (Fig. S8a). Correspondingly, a significant enrichment of ABC variants was obviously observed among CRC risk loci compared with non-ABC variants (Fig. 3i). Additionally, we found GWAS signals were most significantly enriched among ABC enhancers compared with other cis-regulatory elements that predicted from ATAC peaks, H3K27ac peaks, FANTOM5 enhancers or eQTLs in CRC (Fig. 3j). Moreover, we

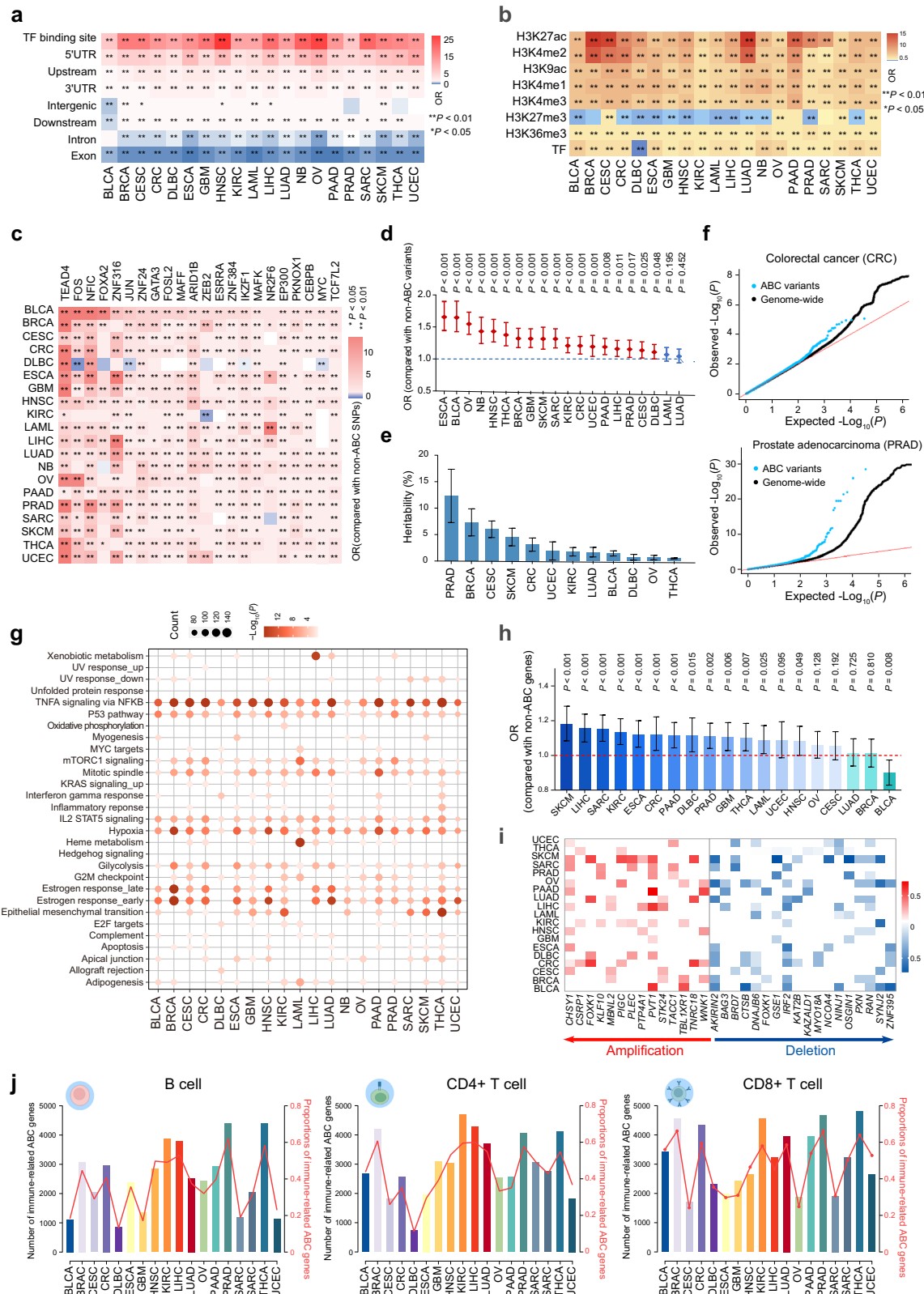

performed in-depth analysis using LDSC to assess the proportion of ABC variants associated with heritability for CRC, and found that ABC variants could explain a significant fraction (1.66%) of cancer heritability (Fig. S8b). We further observed that variants with the higher ABC score were more enriched in the eQTL datasets with different *P* value cutoffs, highlighting the ability of ABC model in dissecting potential functional regulatory variants (Fig. S9). Similarly, these ABC variants also exhibited more significant population-associated *P* values

than genome-wide variants (Fig. 3k). These in-depth analyses provide solid evidence that ABC variants could explain a considerable proportion of CRC heritability.

## The ABC variant rs4810856 at 20q13.13 is associated with CRC risk

To link the regulatory variants located in ABC enhancers to CRC susceptibility, we performed joint analysis of ABC variants with GECCO

**Fig. 2 | Functional characterization of ABC variants and target genes.**
**a** Enrichment analyses of ABC variants in each functional category of genomic distribution compared with control variants (non-ABC variants). *P*-values were calculated by two-tailed Fisher's exact test. **b** Enrichment analyses of ABC variants in regulatory elements including H3K27ac, H3K4me2, H3K9ac, H3K4me1, H3K4me3, H3K27me3, H3K36me3, TF-binding sites compared with non-ABC variants. *P*-values were calculated by two-tailed Fisher's exact test. **c** Heatmap of enrichment analyses for ABC variants within TF-binding sites from ChIP-seq data of ENCODE portal. *P*-values were calculated by two-tailed Fisher's exact test. **d** Enrichment analyses of ABC variants in cancer-related GWAS variants (LD ≥ 0.2) compared with non-ABC variants. *P*-values were calculated by two-tailed Fisher's exact test and bars indicate 95% CIs. **e** Proportion of GWAS trait heritability explained by ABC variants. The error bars represented standard error. **f** Quantile-quantile (QQ) plots of *P* values from GWAS of selected traits. ABC variants were shown in comparison with genome wide variants. GWAS variants were binary annotated using ABC variants with *P* < 0.05. **g** GSEA enrichment analyses of ABC genes in MsigDB hallmark gene sets. The circle color represents the significance of enrichment, and the circle size denotes the number of ABC genes within each gene set. **h** Enrichment analyses of ABC genes in somatically mutated genes from COSMIC database compared with non-ABC genes. *P*-values were calculated by two-tailed Fisher's exact test and bars indicate 95% CIs. **i** The frequencies of amplification and deletion (red indicated amplifications and blue indicated deletions) in ABC genes. Genes recurrently mutated among 20 cancer types were shown. **j** The number of ABC genes associated with immune cell infiltration estimated by TIMER in each cancer type (left y-axis). The right y-axis shows the proportion of these immune-related ABC genes. The cell symbols were created with BioRender.com. Source data are provided with this paper.

GWAS data, which included 17,789 CRC cases and 19,951 healthy controls (Supplementary Table 3). We totally identified 4847 variants showing evidence of association with CRC risk ($P < 0.05$) (Fig. 4a). Of those, the regulatory variant rs4810856 at 20q13.13 prioritized the most potential with the highest ABC score, which was connected to three target genes: *PREX1*, *CSE1L* and *STAU1* among ABC enhancer-gene regulatory maps (Fig. 4b). To validate the association between rs4810856 and CRC risk, we tested the risk effect of rs4810856 in multiple CRC European population cohorts, including the meta data (OR = 1.03, $P = 3.02 \times 10^{-2}$), BioBank Japan Project (BBJ) (OR = 1.08, $P = 1.18 \times 10^{-4}$), UK Biobank (OR = 1.06, $P = 1.02 \times 10^{-2}$) and GECCO (OR = 1.04, $P = 5.59 \times 10^{-3}$) (Supplementary Data 2 and 3). Furthermore, we conducted a two-stage case-control study consisting of 6024 cases and 10,022 controls in Chinese population and the demographic characteristics were detailed in our previous research[24]. The ABC variant rs4810856-C allele conferred a consistent genetic predisposition to CRC in both stages, with adjustment for gender, age group, smoking status and drinking status, respectively. In coherence with the results from public datasets, we combined the results from the two stages and observed that the rs4810856-C allele carriers contribute to an increased risk of CRC compared with T allele carriers (OR = 1.11, 95% CI = 1.04–1.16, $P = 4.02 \times 10^{-5}$) in additive model (Table 1).

## The ABC variant rs4810856 acts as an allele-specific enhancer to promote *PREX1*, *CSE1L* and *STAU1* expression

To determine the ABC variant rs4810856 might function as the causal SNP with a regulatory effect in CRC, we performed functional annotation of this variant rs4810856 by interrogating Cistrome epigenome functional data alongside our ATAC-seq and ChIP-seq data. We observed that the target region harboring rs4810856 was enriched within active histone modification peaks (H3K4me1 and H3K27ac) and open chromatin accessibility (DNase-seq and ATAC-seq peaks, suggesting this region presented the active enhancer property (Fig. S10a). Interestingly, the rs4810856 have significant eQTLs with *PREX1*, *CSE1L* and *STAU1* in our own CRC samples, as C-allele carriers present higher expression of three target genes, compared to T-allele carriers (Fig. 4c). To test whether there were long-range chromatin interactions between enhancer containing rs4810856 and three target gene promoters, we plotted the TAD structure in rs4810856-centered window from Hi-C data of HCT116 cells. The results showed that rs4810856 harboring enhancer and promoters of three target genes were present in the same TAD, indicating that the chromatin interactions between enhancer and promoters might occur to distally affect target genes expression (Fig. 4d). Moreover, the dual-luciferase reporter assays also showed that rs4810856-C allele had higher enhancer activity than that with T allele in both SW480 and HCT116 cells (Fig. 4e, f). Taken together, these results reveal that the ABC variant rs4810856 C-allele could elevate the enhancer activity to promote *PREX1*, *CSE1L* and *STAU1* expression.

## ZEB1 preferentially binds to the risk allele of rs4810856 to affect target genes expression

Transcription factors (TFs) are involved in the precise regulation of gene expression and implicated in cancer pathogenesis[24,25]. Given that allele-specific activity of variants in enhancers can be triggered by binding to specific TFs[19], we first performed TF motif analysis using JASPAR and identified ZEB1 as a candidate factor that preferentially binds to the C allele of rs4810856 (Fig. 4g). Moreover, ChIP-seq data from the Cistrome database also indicated that ZEB1 maps within the region surrounding variant rs4810856 in CRC RKO cell line (Fig. S10a). In addition, we further experimentally validated the binding of ZEB1 to this region by electrophoretic mobility shift assays (EMSA) and observed that the rs4810856[C] allele exhibited a preferential binding to nuclear extracts compared with the rs4810856[T] allele in CRC SW480 and HCT116 cells (Figs. 4h and S10b). Intriguingly, super-shift EMSA also indicated that the binding signals at rs4810856-C allele is gradually reduced with the increasing amount of ZEB1 antibody in a dose-response manner (Figs. 4i and S10c). Meanwhile, the ChIP-qPCR results also showed that the binding peaks of ZEB1 overlapped the enhancer region containing rs4810856, and the binding of ZEB1 to this region was more statistically significant in SW480 cells carrying rs4810856[CC] than in HCT116 cells carrying rs4810856[CT] (Figs. 4j and S10d). Moreover, when we knockdown ZEB1 in SW480 and HCT116 cells at an increasing dose, the differences in luciferase activity between both alleles of rs4810856 was significantly attenuated in a dose-dependent manner (Figs. 4k and S10e). In contrast, the differences in luciferase activity between both alleles of rs4810856 was enhanced in a dose-dependent manner when we overexpressed ZEB1 at an increasing dose, suggesting that ZEB1 preferentially bound to the risk allele of rs4810856 in an allele-specific manner (Figs. 4l and S10f). Notably, the expression of *ZEB1* was found to be positively correlated with the expression of *PREX1*, *CSE1L* and *STAU1* in both TCGA CRC samples and our CRC tissues, and the positive correlations were more significant in carriers with rs4810856-C allele, compared to carries with T allele (Fig. S11). Altogether, these findings demonstrated that ZEB1 preferentially binds to the risk allele of rs4810856 to promote the expression of *PREX1*, *CSE1L* and *STAU1*.

ZEB1 as a prime element of a network of TFs, plays an essential role in tumorigenesis[26]. The expression of *ZEB1* altered predominantly in various cancer types (Fig. S12a). Particularly, we found *ZEB1* was highly expressed in CRC tumor tissues compared with adjacent normal tissues in two independent datasets from GEO database (Fig. S12b). To investigate the biological roles of *ZEB1* in CRC progression, we performed CCK-8 assays in SW480 and HCT116 cells and found that knocking down of *ZEB1* markedly inhibited the proliferation in both cells (Fig. S12c). This findings were consistent with the data from genome-wide CRISPR/Cas9-based loss-of-function screens of CRC cells, indicating that *ZEB1* might play an essential role in cell proliferation (Fig. S12d).

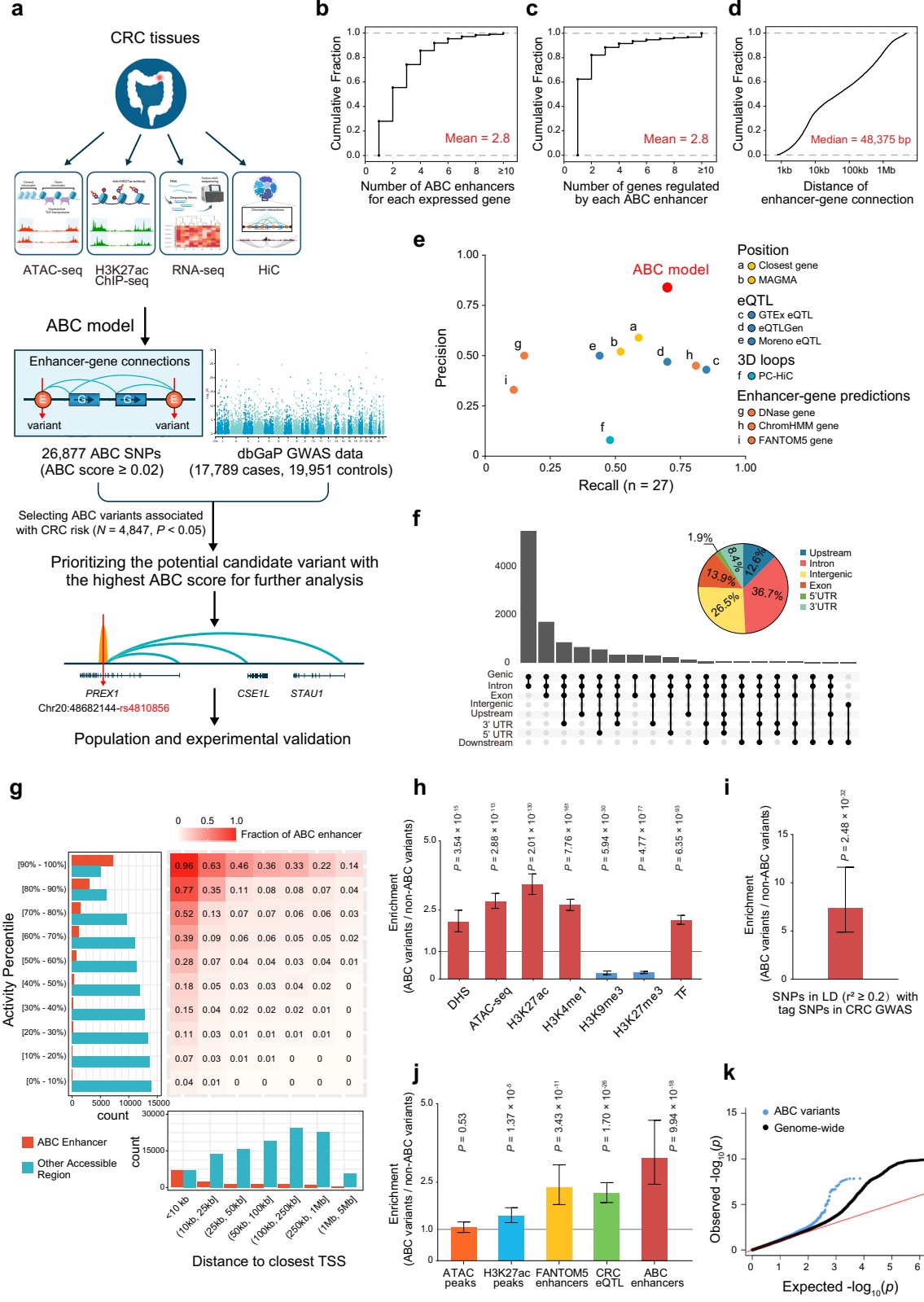

### Direct effects of the ABC variant rs4810856 on ZEB1 affinity, target genes expression and cell proliferation

To further investigate whether rs4810856 is directly involved in ZEB1 binding and the expression of target genes *PREX1*, *CSE1L* and *STAU1*, we applied CRISPR/Cas9-mediated genome editing approach[27], to precisely edit the genotype of rs4810856, where the genotype was successfully converted from CC to CT in SW480 cells, and from CT to TT in HCT116 cells (Fig. 5a, b). We performed ChIP-qPCR assays to validate

the activator role of ZEB1 in vivo, and found that the binding affinity of ZEB1 to the enhancer region harboring rs4810856 is attenuated in mutated cells (SW480[CT] and HCT116[TT]), compared with the parental cells (SW480[CC] and HCT116[CT], Fig. 5c). Additionally, the expression levels of *PREX1*, *CSE1L* and *STAU1* were predominantly decreased in both mutated cells, compared to the parental cells in both SW480 and HCT116 cells (Fig. 5d). Noticeably, the expression differences between both alleles were significantly attenuated upon

**Fig. 3 | Identifying enhancer-gene regulatory connections in our CRC tissues by ABC model. a** The flowchart of ABC model predicting functional variant in CRC tissues, which was created with BioRender.com. **b** Cumulative fractions of the number of enhancers predicted to regulate each gene (mean = 2.8). **c** Cumulative fractions of the number of genes regulated by each ABC enhancer (mean = 2.3). **d** Cumulative fractions of the genomic distances between the enhancer and the gene for each predicted enhancer-gene connection (median=48,375 bp). **e** Precision-recall plot of the accuracy for assigning genes to regulatory variants that predicted by ABC model or other previous predictions, considering a credible set consisting of 27 variant-gene connections which were validated by functional experiments. Recall indicated fraction of the variants identified, and precision indicated fraction of the target genes that was predicted. **f** Genomic annotation of ABC enhancers. Pie chart indicates the proportions of ABC enhancers annotated with each positional category. Upset plot displays the number of ABC enhancers

between groups for each intersection. **g** Summary of ABC enhancers in CRC tissues. Plot included 124,474 non-promoter candidate elements in terms of ATAC peaks. The coloring of the heat map represented the fraction of elements in the corresponding distance and activity bins that are ABC enhancers. **h** Enrichment analyses of ABC variants in regulatory elements compared with non-ABC variants in CRC tissues. *P*-values were calculated by two-tailed Fisher's exact test and bars indicate 95% confidence intervals (CIs). **i** Enrichment analyses of ABC variants in CRC GWAS signals (LD ≥ 0.2). *P*-values were calculated by two-tailed Fisher's exact test and bars indicate 95% confidence intervals (CIs). **j** Enrichment analyses of CRC GWAS signals in ABC enhancers and other regulatory elements. Results were calculated by two-tailed Fisher's exact test. Error bars represented the 95%CIs. **k** Quantile–quantile plot of CRC GWAS *P* values. ABC variants are shown in comparison with genome-wide variants. GWAS variants were binarily annotated using ABC variants. Source data are provided with this paper.

ZEB1 knockdown (Fig. 5e), while the expression differences were significantly increased upon ZEB1 overexpression (Fig. S13a), indicating that rs4810856 acts as an allele-specific enhancer mediated by ZEB1 to directly regulate the expression of three target genes.

Considering that the permissible physical interactions in chromatin are essential for enhancers in regulating gene expression, we then conducted chromosome conformation capture (3 C) assays to examine the existence of chromatin interactions between rs4810856 enhancer and its target genes promoters. When anchored at the enhancer region containing rs4810856, three target genes *PREX1*, *CSE1L* and *STAU1* promoters all showed a stronger interaction with the risk variant region than any of other neighboring BssKI sites tested. Notably, the interaction frequencies were decreased in mutated cells compared to the parental cells in both CRC cell lines (Fig. 5f), suggesting that rs4810856 could establish allele-specific long-range chromatin loops with *PREX1*, *CSE1L* or *STAU1* promoters. Moreover, we further evaluated the direct effect of the risk variant on cell proliferation phenotypes and found that the C > T change of rs4810856 markedly suppressed the proliferation rate and colony formation capacity of CRC cells (Fig. 5g, h). Meanwhile, we observed a significant attenuation of the differences in cell proliferation between both alleles upon ZEB1 knockdown, whereas these differences were significantly increased upon ZEB1 overexpression (Fig. S13b and S13c). Collectively, these series of results provided strong and direct evidence that the risk C allele of rs4810856 functions as an allele-specific enhancer to provoke the expression of *PREX1*, *CSE1L* or *STAU1* through ZEB1-mediated long-range enhancer-promoter interactions, ultimately contributing to the proliferation of CRC cells.

### PREX1, CSE1L and STAU1 play synergistic effects in activating p-AKT signaling to provoke CRC cells proliferation

To investigate the potential roles of *PREX1*, *CSE1L* and *STAU1* in CRC pathogenesis, we first compared the expression of these three genes in tumor and adjacent normal tissues in GEO and our own CRC tissues. Results showed that *PREX1*, *CSE1L* and *STAU1* were significantly overexpressed in tumor tissues than in normal tissues in both cohorts (Fig. S14). We next examined the effects of these three genes on cell phenotypes and found that the overexpression of *PREX1*, *CSE1L* and *STAU1* substantially increase the cell proliferation of SW480 and HCT116 cells, respectively (Figs. 6a and S15a). Expectedly, these results were further validated in colony formation assays (Figs. 6b and S15b). In accordance with these results, the data from genome-wide CRISPR/Cas9-based loss-of-function screens also indicated that *PREX1*, *CSE1L* and *STAU1* are essential for cell proliferation in CRC COLO678 cell line (Fig. 6c). To assess the synergistic effect in vivo, we established cohorts of mice-bearing tumor xenografts driven by CRC cells. In accordance with results in cell lines, the results showed that the growth of xenograft tumors with *PREX1*, *CSE1L* or *STAU1* overexpression is substantially increased, compared with that of control tumors. Interestingly, the group within overexpression of three genes *PREX1*, *CSE1L* and *STAU1*

presented a largest tumor volume among all tested groups (Figs. 6d and S15c). Similarly, H&E staining and immunohistochemical analysis (Ki67, PREX1, CSE1L and STAU1) also support the synergistic effect of three genes on tumor growth (Figs. 6d and S15c). Altogether, these findings further support that *PREX1*, *CSE1L* or *STAU1* can function as potential oncogenes involved in CRC tumorigenesis.

We next sought to dissert the biological pathways underlying the synergistic effects of these three target genes. Previous studies have indicated PREX1 belongs to a family of Rac guanine nucleotide exchange factors (RacGEFs), which is associated with many oncogenic processes including MAPK and AKT pathways[28]. In addition, CSE1L has been suggested function as a key mediator in cellular proliferation and apoptosis, while PI3K/AKT signaling pathway was documented to closely involve in CSE1L-induced tumor progression[29]. For STAU1, which serves as a double-stranded RNA-binding protein, has been widely reported to participate in mRNA degradation via STAU1-mediated mRNA decay (SMD) progress[30]. To recognize the targets of STAU1, we first performed RNA-immunoprecipitation followed by sequencing (RIP-seq) with anti-STAU1 antibody in SW480 cells, and successfully identified 1,341 candidate target mRNAs (Supplementary Data 4). Of these, STAU1 showed stronger binding affinity to *PTEN* mRNA (Fig. 6e, f). We then conducted RIP-coupled qRT-PCR assays and further validated that STAU1 predominantly bind with *PTEN* mRNA in both cancer cells (Fig. 6g). Besides, the overexpression of *STAU1* also greatly decreased the expression level of *PTEN* mRNA in both CRC cells (Fig. 6h). Consistent with this, a negative correlation between the expression of *PTEN* and *STAU1* was observed in both our CRC tissues and GSE9348 dataset (Fig. S16). These results suggested the binding preference of STAU1 to *PTEN* mRNA and induced its mRNA decay.

PTEN is generally known as a dual-specificity protein phosphatase that antagonizes the PI3K/AKT signaling pathway[31], thus it was reasonable to hypothesize that STAU1 might mediate the AKT activation through downregulating the expression of *PTEN*. Given that *PREX1*, *CSE1L*, *STAU1* were all involved in the AKT signaling pathway, we assumed that those genes might exert synergistic effects to activate AKT. To examine this, we constructed *PREX1*, *CSE1L* and *STAU1* overexpressed independently and simultaneously in SW480 and HCT116 cells. The results revealed that cells with *PREX1*, *CSE1L* or *STAU1* overexpression exhibited a higher level of phosphorylated AKT (p-AKT) compared to the control (Fig. 6i). Interestingly, the group with simultaneously overexpression of all three genes presented the highest level of p-AKT among all groups (Fig. 6i). Collectively, these findings demonstrate that PREX1, CSE1L and STAU1 exhibit synergistic effects to facilitate CRC cells proliferation via pathological activation of p-AKT signaling pathway.

## Discussion

Large-scale GWAS have identified thousands of genetic variants associated with human traits and diseases, but over 90% of them locate in non-coding regions, such as enhancers, indicating that they might

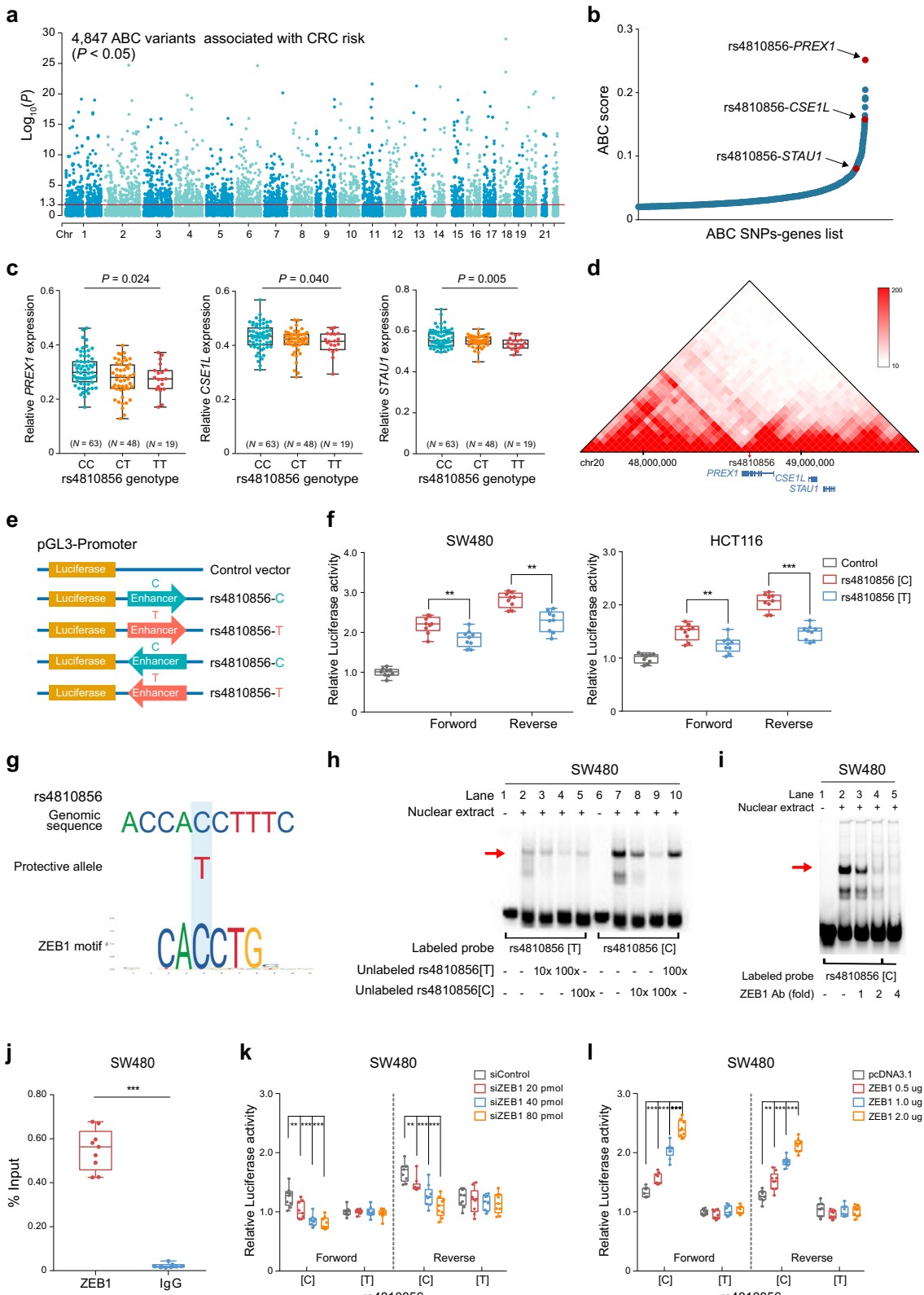

involve in gene expression regulation in proximal or long distances[1,32]. However, mapping non-coding variants to their target genes remains a significant challenge. In this study, we leveraged ABC model to establish genome-wide regulatory maps by integrating large-scale multi-omics data, identifying 544,849 enhancer-gene connections associating 266,956 enhancers and 216,268 target genes across 20 cancer types. Notably, we found only 0.5% of enhancer-gene connections were shared among pairs of cancer types, indicating that most

regulatory landscapes were highly cancer-type-specific, which is consistent with the findings reported by Lijin et al. [33]. This cancer-type specificity can be attributed to the fact that cis-regulatory elements are often cell-type-specific and exhibit activity only in certain cell types. Furthermore, the distinct genetic alterations and signaling pathways in different cancer types also contribute to the cancer-type-specific regulatory landscape[34]. Meanwhile, the systematic characterizations of enhancer-gene pairs emphasize the significance of the ABC mapping

**Fig. 4 | ABC variant rs4810856 acts as an allele-specific enhancer to promote *PREX1*, *CSE1L* and *STAU1* expression. a** Manhattan plot for the associations between ABC variants and CRC risk in GECCO cohort. The *P* values (-log10) of the variants (y-axis) are presented according to chromosomal positions (x-axis, NCBI build 38). *P* < 0.05 was considered statistically significant (denoted by red line). **b** The ABC score of the variant-gene pairs associated with CRC risk. **c** eQTL analyses for the correlation between rs4810856 and the expression of target genes in our own CRC tissues ($N_{CC}$ = 63; $N_{CT}$ = 48; $N_{TT}$ = 19). The gene expression was normalized relative to *GAPDH*. The association between the genotype and gene expression was assessed using linear regression model. Data are shown as the median (minimum to maximum). **d** TAD overlaid with gene annotations surrounding rs4810856 from the Hi-C in HCT116 cells. **e, f** The relative luciferase activity in SW480 and HCT116 cells. Data were shown as the median (minimum to maximum), from three independent experiments with three technical replicates. ***\****P* < 0.0001, *\**\**P* < 0.001 were

calculated by a two-sided Student's *t*-test. **g** The rs4810856 [C] resides within ZEB1 binding motif predicted by JASPAR. **h** EMSAs of rs4810856 in SW480 cells. **i** ZEB1 super-shift EMSAs in SW480 cells. 1×, 2× and 4× represented 0.1 μg, 0.2 μg and 0.4 μg of ZEB1 antibody. **j** Chromatin enrichment of ZEB1 at the rs4810856 in SW480 cells. Data were presented as the median (minimum to maximum) and normalized to the input from three repeated experiments with three replicates. ***\****P* < 0.0001 were calculated by a two-sided Student's *t*-test. The effect of *ZEB1* knockdown (**k**) or overexpression (**l**) on relative luciferase activity of vectors containing rs4810856 [C] or rs4810856 [T] allele in SW480 cells. The center line of the box presentation as the median, box limits indicated upper and lower quartiles, and whiskers indicated the maximum and minimum. ***\****P* < 0.0001, *\**\**P* < 0.001, *\**P* < 0.01 were calculated by a two-sided Student's *t*-test, from three independent experiments with three technical replicates. Source data are provided with this paper.

strategy in prioritizing functional variants and target genes, providing valuable insights into the molecular mechanisms of cancer development. Moreover, ABC model enable link the causal variants to target genes with higher precision than other alternatives, such as distance threshold, eQTL method and other single epigenetic annotation. Furthermore, we identified more than 30,000 enhancer-gene connections in our CRC tissues using ABC approach. Then, we systematically screened ABC variants associated with CRC risk in 17,789 cases and 19,951 controls using chip data in European population and independently validated in a large-scale population consisting of 6024 cases and 10,022 controls in Chinese population. We identified a regulatory variant rs4810856 that is associated with an increased risk of CRC (OR = 1.11, 95%CI = 1.05–1.16, *P* = 4.02 × 10$^{-5}$). Mechanistically, the ABC variant acted as an allele-specific enhancer to distally facilitate expression of *PREX1*, *CSE1L* and *STAU1*, which synergistically activated p-AKT signaling to drive CRC tumorigenesis (Fig. 7).

Genetic risk factors can contribute to the development of cancers, some of which were potentially modifiable[35]. Therefore, it is crucial to understand the molecular targets of known carcinogens in order to design effective therapeutic interventions. we found that the target genes affected by ABC enhancers were closely related to cancer signaling pathways, high mutation burden, immune infiltration, and pharmaceutical targets. These results indicated the importance of the ABC genes which might have significant clinical applications. Numerous studies have demonstrated the critical role of the immune system in cancer development and progression. For instance, immune infiltrates have been identified as an integral component of the tumor microenvironment, and have been shown to play a critical role in tumor progression, responses to immunotherapy and clinical outcomes[36]. In our study, the immune infiltration-associated genes identified by the ABC approach may provide promising therapeutic strategies for cancers. Furthermore, our discovery of target genes involved in cancer signaling pathways, high mutation frequency, and pharmaceutical targets might enable more precise therapeutic targeting of cancers.

So far, a number of approaches have been developed to connect non-coding variants to their target genes. The method simply assigned the closest gene to each variant is widely used to annotate GWAS loci[37]. However, fueled by the advances in 3D genomics, regulatory elements in many cases were found located at great genomic distances from their target genes, or even bypass the closest genes to interact with the distant genes[38]. In addition, eQTL is a common approach to interpret genetic architecture of gene expression[39]. Encouraged by the international projects such as TCGA and GTEx, maps of eQTLs are being built in increasingly large-scale studies in tumor or normal tissues linking regulatory variants to target genes[13,40]. Nevertheless, eQTL is unable open the "black box" between the variants and gene expression as it is analyzed based on statistical algorithms, explaining a limited proportion of disease SNP heritability[41]. With the continuous development of emerging technologies, researchers have begun to

decipher the mechanism of gene expression regulation from an epigenetic perspective, by correlating the activity of enhancers with gene expression or basing on chromatin interaction frequency to predict the target genes of candidate regulatory elements[11,42]. However, its warranted to be noted that both chromatin activity and interaction frequency could cooperatively contribute to gene expression.

The ABC model systematically integrated the enhancer activity from ATAC-seq/DNase-seq and H3K27ac ChIP-seq data, as well as Hi-C-derived contact frequencies to create genome-wide enhancer-gene connections that might enhance the comprehension of linking gene expression regulation to function. Recent study demonstrated that ABC model performed well at classifying regulatory connections and outperformed other methods using a subset of the CRISPR data[15]. In this study, we found that ABC enhancers were more significantly enriched in functionally validated variants compared with other cis-regulatory elements predicted from ATAC peaks, H3K27ac peaks, FANTOM5 enhancers or eQTLs. Meanwhile, ABC model performed better among a credible set consisting of 27 variant-gene pairs that were successfully validated by functional experiments in CRC, with the highest precision in assigning target gene to those variants compared with other approaches such as distance thresholds, eQTL, gene expression correlation, enhancer activity or chromatin interaction frequency. Altogether, our findings highlight the enormous advantages of ABC model linking enhancer variants to target genes strategies, improving the identification of connection between functional variants and target genes in human cancers.

Traditionally, studies often focus on the gene regulation whose variant is only involved a single gene (one-to-one). But recently, increasing evidence have demonstrated that one variant might regulate the expression of multiple target genes (one-to-many), and one gene might also be affected by multiple variants (many to one)[43–45]. The enhancer-gene maps created in this study detected that each enhancer was predicted to affect 2.0 genes, while each gene was predicted to be regulated by 2.5 enhancers, indicating that this one-to-many or many-to-one are prevalent in a genome-wide scale. A salient example of one-to-many regulation pattern is rs72725854-harboring enhancer is observed to regulate the transcription process of *PCAT1*, *PVT1* and *c-MYC* in prostate tumors[46]. Interestingly, consistent with this observation, our study identified a functional non-coding variant rs4810856 could simultaneously regulate *PREX1*, *CSE1L* and *STAU1* in CRC, which is supported with high-confidence evidence of bioinformatics analysis and biological experiments. Additionally, the CRISPR/Cas9-mediated genome editing technology further directly verified rs4810856-C allele could increase the expression level of the three genes, and thus provoke the proliferation rate of CRC cells. Importantly, our findings revealed the underlying biological mechanism that is mediated by ZEB1, which facilitated the folding of the 3D genome and bring rs4810856-harboring enhancer and promoters of three genes into proximity. Taken together, our study elucidates that the risk variant rs4810856, as an allele-specific enhancer, facilitates long-range

**Table 1 | Association analyses between ABC variant rs4810856 and CRC risk in the two-stage and combined samples**

| Variant | | Phase I (N=3046) | | | Phase II (N=13,000) | | | Combined study (N=16,046) | | | MAF |
|---|---|---|---|---|---|---|---|---|---|---|---|
| | | Cases/Controls | OR (95% CI)[a] | P[b] | Cases/Controls | OR (95% CI)[a] | P[b] | Cases/Controls | OR (95% CI)[a] | P[b] | |
| rs4810856 | TT | 185/230 | 1.00 (Ref) | | 561/1,208 | 1.00 (Ref) | | 746/1,438 | 1.00 (Ref) | | 0.35 |
| | CT | 662/669 | 1.26 (0.97–1.63) | 0.087 | 1,919/3,762 | 1.01 (0.90–1.14) | 0.813 | 2,581/4,431 | 1.16 (1.05–1.29) | $4.72 \times 10^{-3}$ | |
| | CC | 677/623 | 1.31 (1.04–1.64) | 0.025 | 2,020/3,530 | 1.38 (1.22–1.56) | $2.09 \times 10^{-7}$ | 2,697/4,153 | 1.32 (1.19–1.46) | $2.55 \times 10^{-7}$ | |
| | Dominant | | 1.14 (1.01–1.26) | $3.93 \times 10^{-2}$ | | 1.12 (1.01–1.26) | 0.034 | | 1.16 (1.05–1.27) | $2.77 \times 10^{-3}$ | |
| | Recessive | | 1.14 (0.97–1.32) | 0.099 | | 1.14 (1.06–1.23) | $5.37 \times 10^{-4}$ | | 1.13 (1.06–1.21) | $2.49 \times 10^{-4}$ | |
| | Additive | | 1.15 (1.04–1.28) | $9.25 \times 10^{-3}$ | | 1.11 (1.05–1.17) | $3.78 \times 10^{-4}$ | | 1.11 (1.05–1.16) | $4.02 \times 10^{-5}$ | |

OR odds ratio, CI confidence interval, MAF minor allele frequency.

[a]ORs and 95% CIs calculation were conducted under assumption that variant alleles were risk alleles.

[b]All P values were calculated by unconditional logistic regression model after adjusting for gender, age group, smoking status and drinking status.

enhancer-promoter interactions mediated by ZEB1 to regulate expression of multiple genes, which reveals a mechanism wherein a single variant regulates multiple tumorigenesis-related genes in CRC.

Our study implicates *PREX1*, *CSE1L* and *STAU1* as the target genes for the nominated causal variant rs4810856. PREX1 and CSE1L have been well demonstrated to involve in phosphoinositide AKT signaling pathway and promote cancer progression[28,29]. It is reported that p-AKT signaling pathway is crucial to many aspects of cell growth, survival and metastasis and is frequently activated to facilitate CRC cells proliferation[47,48]. Furthermore, our study validated that *PREX1* and *CSE1L* are upregulated in CRC tissues and the overexpression of *PREX1* and *CSE1L* significantly promote the p-AKT. In term of another target gene STAU1, which was recognized as a dsRNA-binding protein, has been reported to affect mRNA degradation through SMD progress[30]. We explored that *PTEN* which antagonizes the PI3K/AKT signaling pathway is identified as the target mRNA of *STAU1*. We further proposed that STAU1 might activate p-AKT signaling pathway by promoting mRNA degradation of *PTEN* in CRC cells. Interestingly, we found that *PREX1*, *CSE1L* and *STAU1* are not only individually promote AKT phosphorylation but also exert synergistic effect to activate p-AKT pathway further contribute CRC cell proliferation. Collectively, these findings provided strong evidence that *PREX1*, *CSE1L* and *STAU1* play crucial roles in CRC tumorigenesis via synergistically activating p-AKT signaling pathways, implying that a complex network of multiple causative genes is responsible for tumorigenesis of CRC and hopeful to server as targets with therapeutic potential in future.

There are still some limitations in this study. First, cis-regulatory elements are commonly cell-type-specific, and to fully understand the molecular mechanisms underlying cancer risk, it is desirable to dissect enhancer-gene connections at the cell-type-specific level using single-cell technology. However, due to the current limitations in resources for single-cell omics-data, it is challenging to construct comprehensive regulatory maps at the single-cell level. In addition, the ABC approach was based on computational predictions to identify enhancer-gene connections, and the functional relevance of these connections remains to be experimentally validated. CRISPR-Cas9 genome editing system could be further employed to assess the effect of enhancers on gene expression and support the regulatory connections identified by ABC model. Furthermore, ABC model mainly assess the effect active enhancers on gene expression, and does not capture other cis-regulatory elements such as promoters and silencers, which also play an important role in gene expression. Finally, more environmental and lifestyle factors such as physical activity, dietary factors, the regular use of aspirin or other anti-inflammatory drugs should be adjusted to further improve the association determination.

In summary, this study comprehensively creates genome-wide enhancer-gene regulatory maps across multiple cancer types by integrating multi-omics data that correspond to the activity of candidate regulatory elements and chromatin interaction frequency. These regulatory maps provide a foundational reference for identifying regulatory variants and prioritizing disease genes. It is worth mentioning that ABC model performed better at linking functional regulatory elements to target genes compared with the previous known methods such as distance threshold, eQTL, and other single epigenetic annotation. In addition, we applied ABC strategy to construct enhancer-gene regulatory maps in CRC tissues and identified an ABC variant rs4810856 could contribute to CRC risk by facilitating a long-range chromatin interaction to promote the expression levels of *PREX1*, *CSE1L* and *STAU1*, which synergistically activate p-AKT signaling pathway to drive CRC tumorigenesis. The ABC enhancers-genes maps across multiple cancers not only provide an essential resource for understanding gene regulation and the genetic basis of human cancers, but also might shed light on the biological basis for cancer etiology and provide insight into clinic therapy.

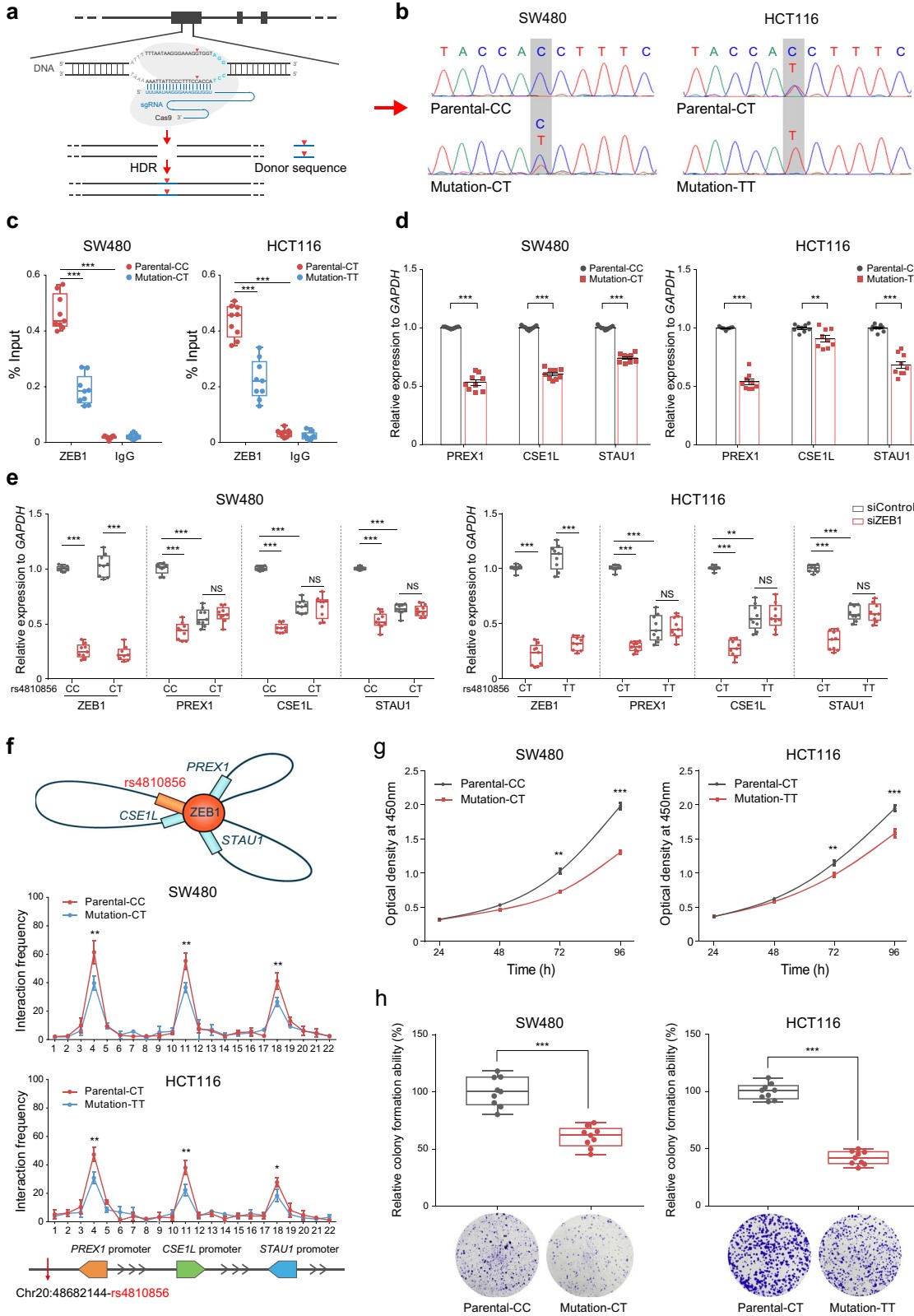

Our study adheres to the Guidelines of the Ministry of Science and Technology (MOST) for the Review and Approval of Human Genetic Resources. This study was approved by the Ethics Committee of Cancer Institute and Hospital and the Ethics Committee of Tongji Hospital, Tongji Medical College of Huazhong University of Science and Technology (HUST). Written informed consent was obtained from each subject, and clinical information was collected from medical records.

### Epigenomic profiling of 20 cancer types

To predict enhancer-gene connections in human cancers, we curated the published epigenomic data including DNase-seq, ATAC-seq, H3K27ac ChIP-seq and HiC from ENCODE, Roadmap and GEO datasets.

**Fig. 5 | Direct effects of the ABC variant rs4810856 on ZEB1 affinity, target genes expression and cell proliferation.** CRISPR/Cas9 mediated single variant mutation of rs4810856 in SW480 and HCT116 cells. The genotype of rs4810856 was converted from CC to CT in SW480 cells (**a**), and from CT to TT in HCT116 cells (**b**). **c** Chromatin enrichment of ZEB1 at the rs4810856 in parental (SW480[CC] and HCT116[CT]) and mutated cells (SW480[CT] and HCT116[TT]). Data were presented as the median (minimum to maximum) and normalized to the input from three repeated experiments, each with three replicates. IgG served as negative control. ***$P < 0.0001$ were calculated by a two-sided Student's $t$-test. **d** Expression of target genes in parental and mutated cells. Data were presented as the mean ± SEM of triplicate experiments, each with 3 technical replicates. ***$P < 0.0001$, **$P < 0.001$ were calculated by a two-sided Student's $t$-test. **e** The effects of *ZEB1* knockdown on target genes expression in parental and mutated cells. The center line of the box presentation as the median, box limits indicated upper and lower quartiles, and whiskers indicated the maximum and minimum. ***$P < 0.0001$, **$P < 0.001$ were

calculated by a two-sided Student's $t$-test, from three independent experiments with three technical replicates. **f** Enrichment quantification of allele-specific 3 C profiles in CRISPR/Cas9 editing cell lines with different rs4810856 genotypes depict the relative interaction frequencies between DNA fragment containing rs4810856 as the anchor and representative BssKI enzyme cutting sites indicated by dot plot. Data were shown as the mean ± SD of triplicate experiments. ***$P < 0.001$, *$P < 0.01$ were calculated by a two-sided Student's $t$-test. **g** The direct effect of rs4810856 genotype on cell proliferation. Results were shown as the means ± SEM from triplicate experiments. ***$P < 0.0001$, **$P < 0.001$ were calculated by a two-sided Student's $t$-test. **h** The direct effect of rs4810856 genotype on colony formation ability. The results presented colony formation ability relative to control cells (set to 100%). Data were shown as the median (minimum to maximum) from triplicate experiments. ***$P < 0.0001$ were calculated by a two-sided Student's $t$-test. Source data are provided with this paper.

The data sources for each cancer type were listed in Supplementary Data 1. In summary, we downloaded BAM files for DNase-seq, ATAC-seq, H3K27ac ChIP-seq from ENCODE[10] and Roadmap datasets (http://egg2.wustl.edu/roadmap/data/byFileType/alignments/consolidated/), and FASTQ files from GEO dataset and processed them using custom pipelines. We aligned reads using BWA[49] removed PCR duplicates using MarkDuplicates function from Picard (http://picard.sourceforge.net), and filtered to uniquely aligning reads using Samtools (https://github.com/samtools/samtools). The FASTQ files of Hi-C was downloaded from GEO dataset and processed them using Juicer tool (https://github.com/aidenlab/juicer). It was previously reported that an averaged Hi-C matrix of several cell lines performed similarly to cell-type matched Hi-C data[14,15]. Thus, we also applied the average Hi-C into analyses for some cells lacking cell-type specific Hi-C and downloaded from website (ftp://ftp.broadinstitute.org/outgoing/lincRNA/average_hic/average_hic.v2.191020.tar.gz). The average Hi-C maps at 5 kb resolution were generated from 10 human cell types and were found to show a strong correlation with cell-type-specific Hi-C maps.

## ABC model construction

We utilized the ABC model (https://github.com/broadinstitute/ABCEnhancer-Gene-Prediction) to predict enhancer-gene connections in each cancer type by integrating multi-omics including chromatin accessibility (ATAC-seq or DNase-seq), histone modifications (H3K27ac ChIP-seq), and chromatin conformation (Hi-C) as previously published described[14,15]. Briefly, the construction of the ABC model involved the following steps using Python (v3.7.0): (1) Identification of peaks on the chromatin accessibility dataset (ATAC-seq or DNase-seq) using MACS2 (v2.1.3) with $P < 0.1$; (2) Counted chromatin accessibility reads in each peak and retained the top 150,000 peaks with the most read counts. We then resized each of these peaks to be 500 bp centered on the peak summit after removing any peaks overlapping blacklisted regions and 500 bp regions centered on all gene TSS (promoters). The resulting set of regions were candidate elements; (3) The element activity was calculated by first counting reads in each candidate element in chromatin accessibility and H3K27ac ChIP-seq experiments, followed by computing the geometric mean of the two assays; (4) By combining activity and Hi-C data, the ABC model computed an ABC score which indicated the regulatory effect of the candidate element for each element-gene pair that is normalized by the product of activity and contact for all other elements within 5 Mb of that gene. A threshold of ABC score ≥0.02 was applied to determine significant gene expression regulatory effects.

## CRC tissues acquisition

In this study, we carried out ATAC-seq, H3K27ac ChIP-seq and RNA-seq in 10 patients with CRC. The clinical characteristics of the 10 CRC patients were provided in Supplementary Table 1. All patients were unrelated Han Chinese hospitalized in hospitals in Wuhan, China during 2020 to 2021. CRC was confirmed by histopathological examination of

surgically removed tumors or biopsy specimens. All patients were not treated with chemotherapy or radiotherapy before tumor resection. Fresh CRC tissues were collected at the time of surgery and analyzed as detailed below. This study was approved by the Ethics Committee of Tongji Hospital, Tongji Medical College of Huazhong University of Science and Technology (HUST). Informed consent was obtained from each patient, the clinical information was collected from medical records. Indirect identifiers (age and gender) have been consented for publication, and the term Gender (indicated in Supplementary Table 1 as Male and Female) was used to indicate the biological attribute.

## ATAC-seq

ATAC assay was performed on CRC tissues by SeqHealth (Wuhan, China). In brief, 500 mg tissue was treated with cell lysis buffer and nucleus was collected by centrifuging for 10 min at 500 g at 4 °C. Transposition and high-throughput DNA sequencing library was carried out by TruePrep DNA Library Prep Kit V2 for Illumina kit (Vazyme, China). The library products were enriched, quantified and finally sequenced on Novaseq 6000 sequencer (Illumina) with PE150 model. Raw sequencing data was first filtered by Trimmomatic (v.0.36), and low-quality reads were discarded and the reads contaminated with adaptor sequences were trimmed. Clean Reads were further treated with FastUniq (v.1.1) to eliminate duplication. Deduplicated reads were then mapped to the human reference genome using bowtie2 (v.2.2.6) with default parameters. Afterwards, we processed the data to generate BAM files with samtools (v.1.12) and made intersect between 10 CRC biosamples with bedtools (v.2.27.1).

## H3K27ac ChIP-seq

ChIP-seq assays were performed with Magna ChIP™ G Tissue Kit (Millipore, USA). The tissue was fixed in 1% formaldehyde for 10 min at room temperature, after which 0.125 M glycine was added and the mixture was sat for 5 min to terminate the crosslinking reaction. The tissue was then treated with cell lysis buffer and nucleus was collected by centrifuging at 800 g for 5 min at 4 °C. Next, nucleus was treated with nucleus lysis buffer and sonicated to fragment chromatin DNA. Antibodies against H3K27ac (10 μg, Abcam, USA) were incubated overnight with the crosslinked protein and DNA for immunoprecipitation reactions with protein A/G magnetic beads. DNA fragments were purified and collected by a Dr.GenTLE Precipitation Carrier kit (Takara, Japan). The purified DNA library was then sequenced on Novaseq 6000 sequencer (Illumina) with PE150 model by SeqHealth (Wuhan, China). We filtered, aligned and processed the data to generate BAM files as described in ATAC-seq.

## RNA-seq

Total RNAs were extracted from CRC tissue using TRIzol (Invitrogen, USA) following the manufacturer's instruction. 2 μg total RNAs were used for stranded RNA sequencing library preparation using Ribo-off

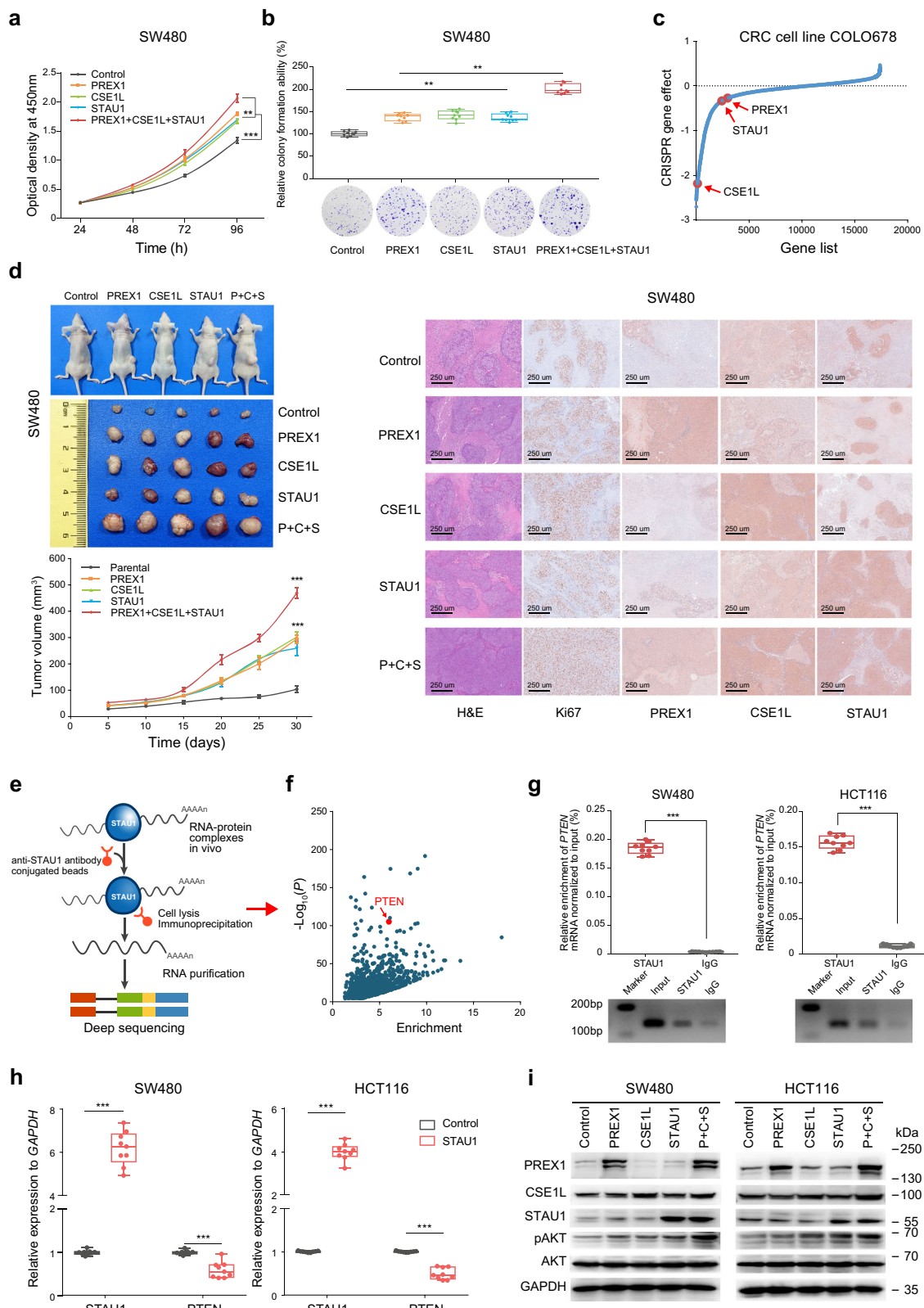

rRNA Depletion Kit (Ribobio, China) and KCTM Stranded mRNA Library Prep Kit for Illumina® (Seqhealth, China). The library products corresponding to 200-500 bps were enriched, quantified and finally sequenced on NovaSeq 6000 sequencer (Illumina) with PE150 model by SeqHealth (Wuhan, China). We filtered, aligned and processed the data to calculate RPKM. Finally, we calculated average gene expression across 10 CRC biosamples.

## Study subjects in association analysis

To study association between ABC variants and CRC risk, we first conducted a GWAS array analysis consisting of 17,789 CRC cases and 19,951 healthy controls. The genotype data was downloaded from the Genetics and Epidemiology of CRC Consortium and Colon Cancer Family Registry (GECCO) program, including phs001078.v1.p1, phs001315.v1.p1 and phs001415.v1.p1. Demographic characteristics

**Fig. 6 | *PREX1*, *CSE1L* and *STAU1* exert synergistic effects to drive CRC development by activating p-AKT signaling pathway. a** The effects of target genes on cell proliferation in SW480 cells. Data were presented as mean values ± SEM from triplicate experiments. ***$P < 0.0001$, **$P < 0.001$ were calculated by a two-sided Student's *t*-test. **b** The effects of target genes on colony formation ability in SW480 cells. The results presented colony formation ability relative to control cells (set to 100%). Data were shown as the median (minimum to maximum) from triplicate experiments. ***$P < 0.0001$ were calculated by a two-sided Student's *t*-test. **c** The potential effects of target genes on proliferation of COLO678 cell line from CRISPR/Cas9-based loss-of-function screens. **d** The effects of target genes on growth of xenograft tumors in nude mice. Representative images, growth curves of xenograft tumors, H&E staining and immunohistochemical analysis derived from lentivirus-mediated SW480 cells were shown. The results were shown as the means ± SD for five mice per group. ***$P < 0.0001$, *$P < 0.01$ were calculated by a two-sided Student's *t*-test. **e** Schematic of the RIP depicting the identification of target mRNAs binding with STAU1. **f** The target mRNAs binding selectively with STAU1 in SW480 cells. MACS2 were used for peak calling to obtain the enrichment fold change and *P* values for the mRNA associated with STAU1 binding. **g** STAU1 RIP coupled with qRT-PCR for *PTEN* mRNA in SW480 and HCT116 cells. Data were presented as the median (minimum to maximum) and normalized to the input from three repeated experiments with three replicates. ***$P < 0.0001$ were calculated by a two-sided Student's *t*-test. **h** The effect of *STAU1* overexpression on *PTEN* expression in SW480 and HCT116 cells. The center line of the box presentation as the median, box limits indicated upper and lower quartiles, and whiskers indicated the maximum and minimum. ***$P < 0.0001$ were calculated by a two-sided Student's *t*-test, from three independent experiments with three technical replicates. **i** The effects of target gene on p-AKT signaling in both CRC cell lines by western blot. Source data are provided with this paper.

were obtained from the previously published study (Supplementary Table 3).

We further conducted a two-stage case-control study to evaluate the associations between candidate variant and CRC risk in Chinese population. The phase I recruited 1524 CRC cases and 1522 controls from cancer hospital of Chinese Academy of Medical Sciences in Beijing, China. The phase II consisted of 4500 cases and 8500 controls from Tongji Hospital of Huazhong University of Science and Technology (HUST), Wuhan, China. All controls were cancer-free individuals selected from a community nutritional survey when patients were recruited and matched to the cases by gender and age (±5 years). The characteristics of the study subjects were described in our previous research[8]. The phase I of case-control study was approved by the Ethics Committee of Cancer Institute and Hospital. The phase II of case-control study was approved by the Ethics Committee of Tongji Hospital, Tongji Medical College of Huazhong University of Science and Technology (HUST). Written informed consent was obtained from each subject and the study was conducted under the approval of the participating hospitals.

## Imputation and quality control for genotype data from GECCO
Imputation was performed using Michigan Imputation Server[50], with Haplotype Reference Consortium r1.1.2016 (HRC) as a reference panel[51]. We merged all batches into a single set after imputation and exclude SNPs according to following criteria[52]: (1) imputation quality <0.4; (2) minor allele frequency (MAF) <1%; (3) deviating from the Hardy-Weinberg equilibrium ($P < 1 \times 10^{-6}$); (4) missing call frequencies >0.02. (5) mapping to locations on sex chromosome. A total of 37,740 individuals with 2,446,560 SNPs were finally included in GECCO GWAS.

## SNP genotyping for our own CRC samples
Genomic DNA was extracted from peripheral blood samples using the Relax Gene Blood DNA System Kit (Tiangen, China) according to the protocol. SNP genotyping was performed using a TaqMan real-time polymerase chain reaction (PCR) assay (Applied Biosystems, CA) without knowledge of the case or control status of the subjects in both stages. Quality control was implemented as previously described[24].

## Cell lines and culture
SW480 and HCT116 cells were obtained from the China Center for Type Culture Collection (Wuhan, China). Cells were cultivated in Dulbecco's modified eagle's medium (DMEM; Gibco, USA) supplemented with 10% fetal bovine serum (FBS; Gibco, USA) and 1% antibiotics (100 U/mL penicillin and 0.1 mg/mL streptomycin) in a humidified atmosphere of 5% $CO_2$ at 37 °C. The SW480 and HCT116 cell lines used in this study were authenticated by short tandem repeat profiling (Applied Biosystems, USA) and tested for the absence of mycoplasma contamination (MycoAlert, USA); the latest date of test was March 1, 2021.

## Construction of plasmids and RNA interference
A total of 1000 bp DNA fragments surrounding the SNP rs4810856 were downloaded from the NCBI database and synthesized by Tsingke Biological Technology (Wuhan, China). The sequence was cloned into the pGL3-promoter vector (Promega, USA). The mutation transcript was generated by site-specific mutagenesis at the rs4810856 site (C > T) and cloned along the same strategy as used for the wide type sequence. The full-length cDNAs of *PREX1*, *CSE1L*, *STAU1* was subcloned into the pcDNA3.1 (+) vector (Invitrogen, USA) by Genewiz Biological Technology (Shuzhou, China). All plasmids then were verified for sequence. For RNA interference, the siRNA oligonucleotides targeting *ZEB1* and non-targeting siRNA were purchased from RiboBio (Guangzhou, China). The target sequences of siRNA are shown in Supplementary Data 5.

## Transient transfections and dual-luciferase reporter assays
For transient transfection assays, cells were seeded in 96 well plates and simultaneously co-transfected with constructed luciferase vector containing either the rs4810856 [C] or [T] allele and the pRL-SV40 Renilla luciferase plasmid (Promega, USA) by using Lipofectamine 3000 (Invitrogen, USA). Luciferase activity was assessed at 36 h post-transfection using the dual-luciferase assay system (Promega, USA). Renilla luciferase and firefly luciferase activities were detected and the relative luciferase activity was calculated to compare the discrepancy between different alleles.

## Electrophoretic mobility shift assays and super-shift EMSA
Complementary DNA oligonucleotides (25 bp) centered on variant rs4810856 alleles were synthesized by Takara (Japan) and labeled with biotin at the 3' end (Supplementary Data 5). Nuclear extracts of SW480 and HCT116 cells were prepared using Nuclear and Cytoplasmic Protein Extraction Kit (Beyotime, Shanghai, China). EMSA was performed with an EMSA/Gel-Shift Kit (Beyotime, China) under the manufacturer's instructions. Additionally, for the competitive binding assay, unlabeled probes were added to the reaction mixtures at a 10-fold or 100-fold excess compared to that of the labeled probes and incubated for 20 min prior to the addition of labeled probes. For super-shift EMSA, 0.1 μg, 0.2 μg and 0.4 μg of ZEB1 antibody (Abcam, ab155249) was added and incubated with the reaction mixtures for 20 min before the addition of labeled probes. The binding products were detected by streptavidin-horseradish peroxidase conjugate according to a Super-Signal West Femto Trial Kit (Thermo Fisher Scientific, USA).

## Lentivirus production and infection
For lentivirus production and transfection, the full-length cDNA of *PREX1*, *CSE1L* and *STAU1* were subcloned into pLVX-EF1a-P2A-Bsd vector by Viraltherapy technologies (Wuhan, China). The pLVX-EF1a-P2A-Bsd empty vector was used as control. Lentivirus was produced in 293 T cells by transfecting target plasmids with X-tremeGENE9 transfection reagent (Roche, USA). Lentivirus-containing supernatant was

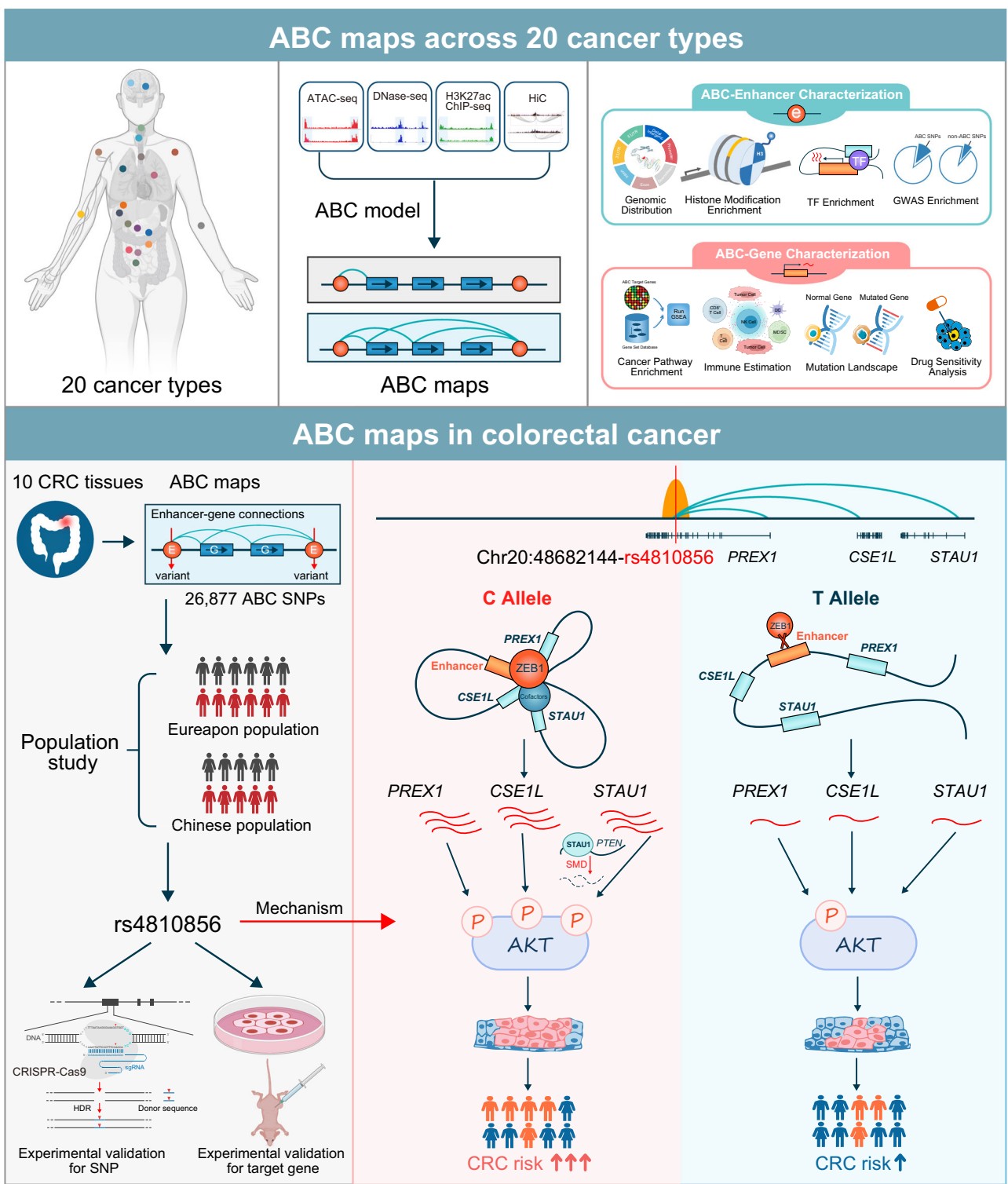

**Fig. 7 | Graphical representation of the regulation mechanism underlying ABC variant rs4810856 and CRC risk.** Firstly, we leveraged ABC model to establish genome-wide regulatory maps genes across 20 cancer types by integrating large-scale multi-omics data including chromatin accessibility (ATAC-seq or DNase-seq), histone modifications (H3K27ac ChIP-seq), and chromatin interaction (Hi-C). We further characterized the functional characteristics of ABC variants based on genomic distribution, functional annotation and GWAS enrichment. Meanwhile, pathway analysis, mutation landscape, immune infiltration and drug response were integrated to the effects of ABC genes in tumorigenesis and clinical application.

Mechanistically, we systematically screened ABC variants associated with CRC risk in 17,789 cases and 19,951 controls using GWAS chip data and independently validated in a large-scale population consisting of 6024 cases and 10,022 controls. We identified an ABC regulatory variant rs4810856 that is significantly associated with an increased risk of CRC (OR = 1.11, 95%CI = 1.05–1.16, $P$ = 4.02×10$^{-5}$). Mechanistically, the ABC variant acted as an allele-specific enhancer mediated by TF ZEB1 to distally facilitate expression of *PREX1*, *CSE1L* and *STAU1*, which synergistically activated p-AKT signaling to drive CRC cell proliferation. The graph was created with BioRender.com.

harvested 48 h after infection and filtered through a 0.45 mm PVDF filter. Finally, lentivirus-containing plasmids were transfected into SW480 and HCT116 cells and ampicillin (100 µg/ml) was added for antibiotic selection. The transfection effect was measured by qRT-PCR (Fig. S17).

## CRISPR/Cas9 mediated single nucleotide mutation

CRISPR/Cas9 mediated single variant mutation at rs4810856 was performed by Genewiz (Suzhou, China). The sgRNAs were designed by using well-established CRISPR design tool (http://crispor.tefor.net/crispor.py). The dsDonor sequence containing the variant rs4810856 (C > T) were designed according to highly efficient sgRNAs. All the oligonucleotides mentioned above are shown in Supplementary Data 5. The sgRNAs was subcloned downstream of the human U6 promoter through BbsI restriction sites in plasmid pSpCas9(BB)-2A-GFP (PX458) (Cat# 48138, Addgene plasmid) and the positive clones were confirmed by sanger sequencing. Transfection was performed in SW480 and HCT116 cells with 80% confluency. 2.5 µg of indicated Cas9 plasmid (PX458-sgRNA) and 1 ug of dsDonor were co-transfected into cells using Lipofectamine 3000. Then 0.8 mg/ml puromycin was added after at 48 h transfection. The remaining live transfected cells were trypsinized and seeded into 96-well plates using fluorescence activated cell sorting (FACS) to establish single cell clone. Finally, the single clones were picked up for subculture and confirmed by sanger sequencing during 14–21 days.

## Quantitative reverse transcription PCR (qRT-PCR)

RNA was extracted from cells using TRIzol reagent (Thermo Fisher Scientific, USA). Reverse transcription was performed by using PrimeScript™ RT-PCR Kit (Takara, Japan) and qPCR was performed in QuantStudio 5 qPCR systems (Applied Biosystems, Thermo Fisher) using SYBR™ Green PCR Master Mix (Takara, Japan). Gene expression was normalized to that of *GAPDH*, which was used as an internal control. All specific primers are listed in Supplementary Data 5.

## Quantitative analysis of chromosome conformation capture assays (3C-qPCR)

3C assays were performed as described in previous study[53] in CRC cell lines carrying different genotypes of rs4810856 that edited by CRISPR/Cas9 system. Cells were fixed with 2% formaldehyde for 10 min and stopped with glycine for 5 min. Next, cells were lysed in lysis buffer and digested with BssKI enzyme (New England Biolabs) at 37 °C overnight. T4 ligase (Thermo Fisher) was added to stop ligation at 16 °C for 5 h. The cross-linked DNA fragments were extracted by phenol/chloroform and precipitated with ethanol. The concentration of 3C DNA samples was measured by qPCR. Additionally, *GAPDH* was used to normalize cell background differences. All 3C-qPCR primers (Supplementary Data 5) were synthesized by Tsingke Biological technology (Wuhan, China).

## Chromatin immunoprecipitation qPCR (ChIP-qPCR)

ChIP assays were performed with ChIP assay kit (Cat# 10086, Millipore, USA). Cells cultured in 15 cm plate were crosslinked with 1% formaldehyde and stopped with glycine. Genomic DNA was extracted from the fixed-chromatin cells and sheared by sonication. 10 µg of antibodies against ZEB1 (Abcam, ab155249) or nonspecific rabbit IgG (Santa Cruz, sc-66931) were incubated overnight with the crosslinked protein/DNA for immunoprecipitation using protein A/G magnetic beads. DNA fragments were purified and collected by a Dr.GenTLE Precipitation Carrier kit (Takara, Japan). The purified DNA library was then detected by qPCR and the primers used in ChIP-qPCR are shown in Supplementary Data 5.

## Cell proliferation assays and colony formation assays

Cell viability was measured using Cell Counting kit-8 (Dojindo, Japan). Cells were seeded in 96-well plates with each well containing 2500 cells in 100 µl of single cell suspension. After a certain time in culture, cells were incubated with 10 µl CCK-8 for a 2 h at 37 °C, and the absorbance was measured at 450 nm using a scanning microplate reader. For colony formation assays, cells were seeded at a density of 5000 cells per well in 6-well cell culture plates. The DMEM with 10% FBS medium was changed every 3 days. After incubation for 12 days, cells were fixed with 100% methanol and then stained with crystal violet solution (Solarbio, Beijing, China) for 20 min. The colonies were then captured and manually counted.

## Western blot

Total protein was harvested from cells using RIPA lysis buffer (Beyotime, China) supplemented with protease inhibitors PMSF (Beyotime, China), cOmplete cocktail (Sigma, USA) and PhosSTOP (Sigma, USA). Protein was quantified using BCA reaction (Beyotime, China) and denatured at 99 °C for 5 min. Equal amount of protein was separated by electrophoresis in 8% SDS-PAGE gels and transferred onto 0.45 mm PVDF. Protein was incubated with antibodies against PREX1(1:1000, CST, Cat# 13168), CSE1L (1:1000, Proteintech, Cat# 22219-1-AP), STAU1 (1:1000, Proteintech, Cat# 14225-1-AP), p-AKT (Ser473, 1:1000, Proteintech, Cat# 66444-1-lg), AKT (1:1000, Proteintech, Cat# 10176-1-AP) and GAPDH (1:1,000, Proteintech, Cat# 60004-1-lg) at 4 °C overnight. HRP-conjugated anti-mouse IgG or anti-rabbit IgG (1:5,000, Proteintech, Cat# SA00001-1, Cat# SA00001-2, respectively) was used as secondary antibody. Chemiluminescence signal was developed with SuperSignal West Pico PLUS Chemiluminescent Substrate (Thermo Fisher Scientific, USA) by Image Lab software. The detail information of antibodies was provided in the Reporting summary.

## RNA immunoprecipitation assays (RIP)

RIP assays were performed using RNA Immunoprecipitation Kit (Cat# P0102, Geneseed, China) according to the manufacturer's recommendations. 5 µg of STAU1 antibody (Proteintech, Cat# 14225-1-AP) or IgG (Santa Cruz, sc-66931) was incubated with magnetic beads at 4 °C for 2 h with rotation. Additionally, cells cultured in 15 cm plate were washed twice with cold PBS and harvested by scraping into 10 mL PBS. Cells were then lysed and collected by centrifuging at 1000 *g* for 5 min at 4 °C. Lysate was incubated with magnetic beads binding with antibodies at 4 °C for 2 h with rotation. After incubation, the beads were washed twice with wash buffer and RNA was purified by using purification columns. In addition, total RNA (input control) was assayed simultaneously. Finally, the coprecipitated RNAs were sequenced by Novogene (Beijing, China) and target RNA was detected by qRT-PCR.

## Xenograft growth of CRC cells in nude mice

Female BALB/c nude mice, aged 4-5 weeks, were purchased from the Vital River Laboratory Animal Technology (Beijing, China), and were allowed to acclimate to local conditions for one week. All mice maintained in the specific pathogen-free (SPF) room under controlled temperature ($23 \pm 3$ °C), and humidity (40–60%) conditions with 12/12 h light/ dark cycle with food and water provided ad libitum. For the xenograft tumor growth assay, nude mice (five per group) were injected subcutaneously in the rear flank with 0.1 ml of cell suspension containing $1 \times 10^7$ cells. When a tumor was palpable, it was measured every five days and its volume was calculated according to the following formula volume = 0.5 × length × width2. Tumor tissue was fixed with paraformaldehyde and was then subjected to hematoxylin and eosin (H&E) staining and immunohistochemical analysis of Ki67, PREX1, CSE1L and STAU1 protein. All experimental procedures were performed in accordance with the relevant institutional and national guidelines and approved by the experimental animal center of Wuhan university. The humane endpoints for mouse experiments encompass a rapid weight loss exceeding 20%, inability to eat and drink, loss of consciousness, severe dehydration, and tumor diameter exceeding 15 mm. None of the tumor sizes in our experiments exceeded 15 mm.

## Statistical analysis

The demographic characteristics between cases and controls were appraised by two-sided $\chi^2$ test or the Student's t- test. The associations between variants and CRC risk were estimated as the odds ratios (ORs) and 95% confidence intervals (CIs) by the unconditional multivariate logistic regression with adjustments for gender, age group, smoking status, and drinking status. Multiple genetic models, such as allelic, dominant, recessive and additive genetic model, were applied to assess the genetic susceptibility of variants to CRC, respectively. For functional assays, figure legends denoted the statistical details of experiments including the statistical tests used, the numbers of replicates, and the data presentation type in relevant figures. All statistical analyses were performed by R (v.3.5.3) or PLINK (v.1.9) software. $P < 0.05$ was considered statistically significant.

## Reporting summary

Further information on research design is available in the Nature Portfolio Reporting Summary linked to this article.

## Data availability

All data relevant to the study are included in the article or uploaded as online Supplementary information. The sequencing data, including ATAC-seq, H3K27ac-seq, and RNA-seq data from 10 CRC tissues that used in this study have been deposited in the Gene Expression Omnibus under accession number GSE222770, which contains RAW sequencing data. The DNase-seq, ATAC-seq, H3K27ac ChIP-seq and Hi-C for 20 cancer type were obtained from ENCODE, Roadmap and GEO datasets, and the data sources were listed in Supplementary Data 1. The summary GWAS statistics for each cancer type are downloaded from the GWAS catalog [https://www.ebi.ac.uk/gwas/docs/file-downloads]. The data that used for association analysis of this study are available by application from the participating consortia, including GECCO [https://www.ncbi.nlm.nih.gov/gap/] and UK Biobank [https://biobank.ctsu.ox.ac.uk/], and the dbGaP accession for GECCO program includes phs001078.v1.p1 [https://www.ncbi.nlm.nih.gov/projects/gap/cgi-bin/study.cgi?study_id=phs001078.v1.p1], phs001315.v1.p1 [https://www.ncbi.nlm.nih.gov/projects/gap/cgi-bin/study.cgi?study_id=phs001315.v1.p1] and phs001415.v1.p1 [https://www.ncbi.nlm.nih.gov/projects/gap/cgi-bin/study.cgi?study_id=phs001415.v1.p1]. The summary GWAS statistics of the BioBank Japan Project (BBJ) was available from [http://jenger.riken.jp/result]. The summary GWAS statistics for the meta-analysis in CORECT, CFR1, MECC1, and GECCO can be obtained from the published research[54]. The Chinese population for association analysis were described in our previous research[8]. Source data are provided with this paper Source data are provided with this paper.

## Code availability

Only publicly available tools were used in data analysis and the parameters have been described wherever relevant in Methods and Reporting Summary.

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

## Acknowledgements

We are grateful to all the study participants, research staff, and students who participated in this work, especially the blood sample donors. This work was supported by Distinguished Young Scholars of China (NSFC-81925032), Key Program of National Natural Science Foundation of China (NSFC-82130098), the Fundamental Research Funds for the Central Universities (2042022rc0026, 2042023kf1005) and Knowledge Innovation Program of Wuhan (2023020201010060) for X.M. The National Science Fund for Excellent Young Scholars (NSFC-82322058), Program of National Natural Science Foundation of China (NSFC-82103929, NSFC-82273713), Young Elite Scientists Sponsorship Program by CAST (2022QNRC001), National Science Fund for Distinguished Young Scholars of Hubei Province of China (2023AFA046), Fundamental Research Funds for the Central Universities (WHU:2042022kf1205) and Knowledge Innovation Program of Wuhan (whkxjsj011, 2023020201010073) for J.T. Youth Program of National Natural Science Foundation of China (NSFC-82003547), Program of Health Commission of Hubei Province (WJ2023M045) and Fundamental Research Funds for the Central Universities (WHU: 2042022kf1031) for Y.Z.

## Author contributions

J.T. and X.M. were the overall principal investigators in this study who conceived the study and obtained financial support. Y.Z., J.T. and X.M. were responsible for the study design and supervised the entire study. P.Y., C.C. and Z.L. performed statistical analyses, interpreted the results and drafted the initial manuscript. B.L., S.C., M.Z., Y.C., F.Z., J.H., L.F., C.N., Y. Li., W.W., H.G., Y.Liu and W.T. performed laboratory experiments. Z.Y., J.L., C.H., X.Y., B.X., H.L., X.Z., N.L. and Y.W. performed data curation and investigation. J.T. and X.M. were responsible for patient recruitment and sample preparation. All authors approved the final report for publication.

## Competing interests
The authors declare no competing interests.
