## [Peer Review File · Nature Communications]

REVIEWER COMMENTS

Reviewer #1 (Remarks to the Author): Expert in CRC genetics, genomics, epigenomics, Hi-C analysis, and eQTLs

The authors state that they undertaken a study demonstrating that genome-wide enhancer-gene regulatory maps link causal variants to target genes underlying human cancer risk.

I have multiple issues with the paper:

1. First and foremost aside from the fact that its rather long the text is all over the place and the experiements. Its really unclear what the salient findings are.
2. In essence the study can be boiled down to an ABC analysis of GWAS data using in house and publicly accessible data to deconvolute the functional basis of GWAS signals. After which the authors conduct some focused functional assays.
3. It is stated that their data demonstrates that the functional variants underlying GWAS risk associations is largely enshrined in non-coding changes. No one would dispute this as its widely established and has been the rationale for deploying Hi-C, PC-HiC and micro-C etc, in combination with eQTL analyses to link risk variants to effector genes.
4. They present a plethora of data on genome-wide enhancer-gene interactions across 20 cancers in what can only be asserted is an unstructured set of analyses.
5. Finally, they seek to decipher the association analyses between ABC variant rs4810856 and CRC risk. The rationale for focusing on this aside from any other cancer is odd to say the least, especially as the association is not statistically significant.

Reviewer #2 (Remarks to the Author): Expert in gastrointestinal cancer genetics, GWAS, QTLs, and genetic epidemiology

The manuscript from Ying et al. described an innovative work on prediction of enhancer-gene connections by using ABC model to build genome-wide regulatory maps in human cancers. They discovered thousands enhancer-gene connections with tissue-specific across 20 cancer types, providing key insights into assigning regulatory elements to potential target genes. Then the authors verified that ABC could perform better in distinguishing causal variants among many possible variants compared with other alternative approaches. Furthermore, through integrating large-scale population studies and a series of biochemical experiments, the authors demonstrated an ABC variant rs4810856-C that acts as

an allele-specific enhancer to distantly influence expression of PREX1, CSE1L and STAU1 in a long-range chromatin interaction manner and mediated by transcription factor ZEB1, resulting in an increased risk of CRC. The manuscript is methodologically sound and the strength of this study is the independent replication with different population. However, the following issues, if addressed, may further support their conclusions.

Major comments

1. The authors performed enrichment analyses among CRC GWAS loci and stated that ABC variants significant enriched in GWAS region with high OR, which suggests ABC variants play critical roles in susceptibility to CRC. Here I think it would be interesting to determine whether ABC variants might function in precision prevention and early detection of CRC?
2. The authors compared the ABC model with other functional annotations (closest gene, eQTL gene, PC-HiC, etc) in the accuracy for assigning genes to regulatory variants (Figure 3e) and concluded that the ABC model had better recall accuracy. It is interesting for the authors to compare the ABC model with the reported functional annotation models, such as INQUIST (Nature 2017. 551(7678):92-94), IMPACT (Nat Genet 2020. 52(12):1346-1354) and PAINTOR with PageRank (Nat Genet 2021. 53(3):392-402).
3. The authors have verified ABC variant rs4810856 acts as the causal variant in CRC tumorigenesis by two-stage population study and multipronged experiments. The authors could try to provide additional evidence to support the significant association by other independent populations?
4. Super-shift EMSA assay conducted in figure 4i indicated that ZEB1 might be the candidate TF which preferentially binds to rs4810856-C allele. However, no specific lines were observed in the figure to represent the enrichment for this TF. An additional ChIP assay of ZEB1 might be helpful to support this issue.
5. The authors have performed dual-luciferase reporter assays to claim the potential regulatory function of ABC variant rs4810856. Luciferase activity between both alleles of this variant was significantly attenuated in a dose-dependent manner upon TF knockdown, which indicates that ZEB1 might involve in the precise regulation of gene expression. Meanwhile, similar results were also observed in the expression of three target genes. I think it would be nice to prove the effects of luciferase activity and the regulatory function of target genes expression based on overexpression of ZEB1.
6. The authors have identified over 30,000 enhancer-gene connections in CRC tissues by using ABC model, whereas only one of which was validated by large-scale population and experiments with high confidence. The rest of enhancer-gene maps detected by ABC model are encouraged to test the effect of enhancers on gene expression in a high throughput technology.

7. A body of comprehensive results in the manuscript indicated that variants located in ABC enhancers were enriched in the regulatory region such as transcription factor binding sites and histone modification for active markers. The authors should provide the information of the TF and histone modification that they included in the analyses in detail.

Minor comments

1. Line 139 “we generated the control variants set (non-ABC variants) with the distribution of MAF and location matched to the set of ABC variants per cancer type”. The authors should state the criterion to generate the non-ABC variants detailly.

2. The authors conducted quality control for genotype data from GECCO according to following criteria: (1) imputation quality < 0.4 ; (2) minor allele frequency (MAF) $< 1\%$; (3) deviating from the Hardy-Weinberg equilibrium ($P < 1 \times 10^{-6}$); (4) missing call frequencies > 0.02 . (5) mapping to locations on sex chromosome. Please explain why chose this threshold as the criterion of quality control or add a reference.

3. Please label the MAF of rs4810856 C/T allele in the table footnotes in Table 1.

4. In the legend of Figure S2 “The circle color indicates the Pearson correlation, and circle size indicates FDR”. FDR abbreviation is not defined.

5. Although the writing is fluent, many sentences in the manuscript are quite long, the authors are suggested shorten their sentences to improve the readability of the text.

Reviewer #3 (Remarks to the Author): Expert in CRCr genetics, GWAS, QTLs, and genetic etiology

In this study, the authors created enhancer-gene maps across 20 cancer types through integrating multi-omics data applying ABC model, providing an informative enhancer-gene connection resource and a novel view of the regulation mechanisms linking non-coding variants that control gene expression to CRC risk. Besides, the variants positioned within ABC enhancers played key roles in tumorigenesis as cancer drivers. Furthermore, they found that target genes affected by ABC variants are closely involved

in multiple cancer pathways, high mutation burden, immune cell infiltration and pharmaceutical targets, which will be helpful for suggestive of potential use in clinical practices. They demonstrated an ABC variant rs4810856-C allele contributes to an increased risk of CRC in large-scale multiracial population cohorts. Mechanically, rs4810856 acted as an allele-specific enhancer to distally facilitate expression of PREX1, CSE1L and STAU1, which synergistically activated p-AKT signaling to drive CRC tumorigenesis. This study provides a valuable approach to elucidate the contribution of genetic variants to various traits and diseases, including cancers. The paper is well performed and the data and experimental results shown supports the author's conclusions. I have the following comments.

Major issues

1. A body of evidence indicated that variants located in ABC enhancers were enriched in the regulatory region within open chromatin status. For example, ABC variants recognized in CRC were remarkably enriched among TF binding sites and active epigenetic markers such as H3K27ac, H3K4me1, ATAC peaks and DNaseI hypersensitive sites (DHSs). I am wondering whether the authors could try to analyze the enrichment of ABC variants among the chromatin state?
2. The results presented that ABC variants were significantly enriched in CRC GWAS loci and ABC genes were associated with the clinical utility including multiple cancer pathways, high mutation burden, immune cell infiltration and pharmaceutical target, highlighting the importance of the connections predicted by ABC model. Accordingly, I am interested to see whether ABC variants might serve as an effective tool for early screening of CRC?
3. The authors have conducted the heritability analyses of ABC variants across 20 cell types, including CRC cells, "ABC variants could explain a significant fraction of cancer heritability, ranging from 0.5% in thyroid carcinoma (THCA) to 12% in prostate adenocarcinoma". Authors can analyze the heritability of ABC variants derived from CRC tissues.
4. The authors performed the dual-luciferase reporter assays and EMSA assay to verify that ZEB1 might be the candidate TF which preferentially binds to the risk allele of rs4810856. To furthermore assure that ZEB1 is the key regulatory factor of the process that rs4810856 influences the expression of three oncogenes, authors are encouraged to perform a ChIP assay in both CRC cell lines used in the EMSA assay.
5. The results showed in single variant-CRISPR/Cas9 editing cell lines are very interesting and adequately support the direct effects of the ABC variant rs4810856 on expression of three target genes and cell proliferation, which are mediated by ZEB1. The remarkable effects of TF ZEB1 on the expressions of three target gene were observed under the consideration of ZEB1 knockdown, the effects of ZEB1 overexpression on it are warranted in the single variant-CRISPR/Cas9 editing cell lines.
6. Furthermore, to test the direct effects of the ABC variant rs4810856 on cell proliferation whether are dependent on the ZEB1, the effects of ZEB1 knockdown or overexpression on cell proliferation by CCK8 and colony formation assays are encouraged to conduct in the single variant-CRISPR/Cas9 editing cell lines with different rs4810856 genotypes.

7. It is ingenious that the authors found the critical mRNA PTEN which provided the clues for the associations between STAU1 and p-AKT signaling pathway in RIP assay in Fig 6a and Fig 6b. The authors should provide the list of the potential target mRNA that could bind to STAU1 in the supplementary material.

8. Since CREs are commonly cell-type-specific with the rapid development of scATAC-seq technology, understanding cancer regulatory mechanisms at the cell type-specific level is required to fully interpret the functional impact of risk variants. Could the authors consider that integrate single-cell sequencing data into the ABC model to identify cell-type specific enhancer-gene connections?

Minor issues

1. Line 139 “we generated the control variants set (non-ABC variants) with the distribution of MAF and location matched to the set of ABC variants per cancer type”. The authors should state the criterion to generate the non-ABC variants detailly.

2. Line 796 “Replicability of enhancer-gene pairs across cancer types”. The authors should describe the method that they conducted to this analysis of Fig 1h detailly in the method.

3. Line 878 “Genes PREX1, CSE1L and STAU1 are in the same TAD”. While the statement in the main text is “The results showed that rs4810856 harboring enhancer and three target genes were present in the same TAD”. The authors should refine this description in the figure legend of Fig 4d.

4. The authors should add the types of cell lines or tissues that TF and histone markers downloaded from ENCODE database in the supplementary material.

5. While the manuscript was understandable, the English writing still need some polish.

Reviewer #4 (Remarks to the Author): Expert in CRC genetics, QTLs, genomics, epigenetics, and functional genomics

This is an interesting work and Ying et.al systematically established genome-wide enhancer-gene maps across 20 human cancer types by integrating the multi-omics data using the ABC model. Numerous enhancer-gene connections were identified with potential regulatory functions associated cancer development, which provided the important insights into assigning the non-coding variants to underlying target genes. Notably, they illustrated that ABC model performed better in linking regulatory variants to target genes than other previous approaches. The authors further focused on an ABC variant rs4810856 which was significantly associated with CRC risk, and claimed the functional mechanism using a series of biological assays. Interestingly, they reported a regulatory role of this variant on three target genes which could exert synergistic effects to activate p-AKT signalling. The work presented was of high quality and would be of broad interest, providing regulatory maps linking causal variants to cancer and

indicating a novel mechanism among cancer growth. The paper is well performed. However, there are some results need further development to support the author's conclusions before publication.

major issues

1. The authors used ABC model systemically integrating several multi-omics data to build genome-wide enhancer-gene maps in 20 cancer types, and found the most regulatory landscapes were highly cancer-type-specific in figure 1h. However, it needs to discuss in-depth about the probable reasons of this tumor specific issue.

2. The authors used an average Hi-C data in some cancer types that lacking cell-type specific Hi-C to build enhancer-gene maps. This might raise concern about whether this average Hi-C data were suitable to compute cancer-specific enhancer-gene maps. The authors should explain the rationality of this replacement.

3. The target genes affected by ABC SNPs were found to related to cancer signaling pathways, high mutation burden, frequent amplification and deletion alterations, immune infiltration and pharmaceutical targets, indicating the target genes were quite important in cancer development. It might be more meaningful to discuss the potential clinical application of these target genes in the discussion.

4. Super-shift EMSA assay performed in figure 4h indicated that ZEB1 might be the candidate TF which preferentially binds to the risk allele of rs4810856. However, there was no specific lines in the figure to represent the enrichment for this TF. An additional CHIP assay of ZEB1 might be helpful to support this issue.

5. The authors conducted dual-luciferase reporter assays to claim the potential regulatory function of the variant. Luciferase activity between both alleles of this variant was significantly attenuated in a dose-dependent manner upon TF knockdown, which provided important evidence to support that ZEB1 might involve in the precise regulation of gene expression. Besides, the effects of luciferase activity in this enhancer region based on overexpression of the TF still needed to be assessed.

6. The ABC SNPs were indicated to be enriched in GWAS region, and could explain a considerable proportion of cancer heritability in figure 2d and 2e. This was an interesting finding that links ABC SNPs to cancer risk and represents these SNPs identified in this study might important role in cancer progression. However, it should describe better how these GWAS variants were selected for each cancer types in the method.

7. The authors performed association analysis and found the significant association between the variant and CRC risk in three independent population including European and Chinese ethnic, which provided strong clues to confirm the genetic susceptibility of rs4810856 to CRC. However, only additive model was used to assess the association in European population. Other genetic models including allelic, dominant and recessive still need to be conducted in this population.

8. Smoking and drinking status are collected and adjusted in the association analysis to exclude the confounding of common risk factors for CRC. However, there were many important environmental factors, including physical activity, dietary factors, the regular use of aspirin or other anti-inflammatory drugs involved in CRC development. It might be meaningful including these confounders in logistic regression to assess more accurate association between the candidate variant and CRC risk.

minor issues

1. The authors stated that ABC model performed better in predicting regulatory elements and their target genes compared with other approaches, but it still needs to describe more detail about this model in the method.

2. The frequency of the risk allele of rs4810856 in Chinese population should be provided in the results.

3. The identity of cell lines used and test free of mycoplasma should be stated in the manuscript.

4. The western-blot lines showed in Figure 6E were too narrow. There is a need to reserve more space above and below the lines.

5. While the manuscript was understandable, the English writing still needs some polish.

Reviewer's Comments:

Reviewer #1 (Remarks to the Author)

The authors state that they undertake a study demonstrating that genome-wide enhancer-gene regulatory maps link causal variants to target genes underlying human cancer risk.

I have multiple issues with the paper:

1. First and foremost aside from the fact that its rather long the text is all over the place and the experiments. It's really unclear what the salient findings are.

Response: Sorry for not described appropriately our findings. The major aim of the study is to provide a systematic and quantitative landscape for regulatory maps linking enhancers to their potential target genes in human cancers, not only to interpret GWAS signals. We have shortened and better organized the structure in revised manuscript.

2. In essence the study can be boiled down to an ABC analysis of GWAS data using in house and publicly accessible data to deconvolute the functional basis of GWAS signals. After which the authors conduct some focused functional assays.

Response: Thank you for the thoughtful comment. Actually, our study was to not only use ABC model to dissect GWAS signals using in house and publicly accessible data, but also try to link the identified ABC variants to risk of colorectal cancer in Chinese population, especially with comprehensive functional assays for the highest ABC score.

3. It is stated that their data demonstrates that the functional variants underlying GWAS risk associations are largely enshrined in non-coding changes. No one would dispute this as it is widely established and has been the rationale for deploying Hi-C, PC-HiC and micro-C etc., in combination with eQTL analyses to link risk variants to effector genes.

Response: Thank you for your reminding. We do acknowledge that it has been the rationale using Hi-C, PC-HiC and micro-C etc, in combination with eQTL analyses to link risk variants to target genes. Our ABC model is just a quantitative strategy to combine enhancer activity and 3D contact frequencies to strengthen the aforementioned signals for pinpointing causal variants and target genes. We elaborated the prominent performance of ABC analysis in the revised manuscript.

Result section:

Page 7, line 211, we state that:

“Notably, ABC model that identified 16 variant-gene connections out of 19 cases (70% recall and 84% precision), had higher precision than other approaches, including predictions based on genomic position, eQTLs, three-dimensional contacts or other enhancer-gene predictions (**Fig. 3e**). This analysis demonstrates that ABC regulatory maps can accurately connect fine-mapped variants to target genes, highlighting the performance for regulatory function annotations by using ABC model.”

Precision-recall plot of the accuracy for assigning genes to regulatory variants that predicted by ABC model or other previous predictions, considering a credible set consisting of 27 variant-gene connections which were validated by functional experiments. Data for eQTLs, 3D loops, and other enhancer-gene predictions were obtained from previous studies.

4. They present a plethora of data on genome-wide enhancer-gene interactions across 20 cancers in what can only be asserted is an unstructured set of analyses.

Response: Sorry for the misleading description. We have better organized the structure in revised manuscript. Actually, apart from constructing an informative profile of enhancer-gene connections across 20 cancers, we further characterized their significant roles in tumorigenesis by multiple cancer pathways, mutation burden, immune cell infiltration and pharmaceutical targets separately, highlighting their potentially clinical applications in cancers. The comprehensive enhancer-gene maps present a promising strategy for distinguishing causal variants and target genes, and provide valuable clues for holistic comprehension of the genetic architecture of cancers.

5. Finally, they seek to decipher the association analyses between ABC variant rs4810856 and CRC risk. The rationale for focusing on this aside from any other cancer is odd to say the least, especially as the association is not statistically significant.

Response: Sorry for the misunderstanding. We have addressed these issues and described more precisely our main findings in revised manuscript. As our response to your Comment 1 and 2, to provide a genome-wide landscape for regulatory maps linking causal variants to target genes in cancer risk, we not only used public available data in bioinformatics, but also focused on colorectal cancer in house as a curated example to demonstrate the exactly molecular mechanisms of ABC model, the strategy of which were popular in the genetics studies (Takata A et al., *Nat Commun.* 2017 Feb 27; 8:14519; Walker RL et al., *Cell.* 2020 Apr 16;181(2):484).

Reviewer #2 (Remarks to the Author)

The manuscript from Ying et al. described an innovative work on prediction of enhancer-gene connections by using ABC model to build genome-wide regulatory maps in human cancers. They discovered thousands enhancer-gene connections with tissue-specific across 20 cancer types, providing key insights into assigning regulatory elements to potential target genes. Then the authors verified that ABC could perform better in distinguishing causal variants among many possible variants compared with other alternative approaches. Furthermore, through integrating large-scale population studies and a series of biochemical experiments, the authors demonstrated an ABC variant rs4810856-C that acts as an allele-specific enhancer to distantly influence expression of PREX1, CSE1L and STAU1 in a long-range chromatin interaction manner and mediated by transcription factor ZEB1, resulting in an increased risk of CRC. The manuscript is methodologically sound and the strength of this study is the independent replication with different population. However, the following issues, if addressed, may further support their conclusions.

Response: Thanks for the overall positive comments. We have addressed the concerns in the following responses.

Major comments

1. The authors performed enrichment analyses among CRC GWAS loci and stated that ABC variants significant enriched in GWAS region with high OR, which suggests ABC variants play critical roles in susceptibility to CRC. Here I think it would be interesting to determine whether ABC variants might function in precision prevention and early detection of CRC?

Response: Thanks for the valuable comment. We performed polygenic risk score analyses to estimate the aggregative effect of ABC variants on CRC risk using PRSice in GECCO cohort, and found that participants in the top 20% of the PRS calculated from ABC variants had a 1.36-fold CRC risk (95% CI = 1.27-1.45) compared with those in the bottom 20%, which was higher than the PRS calculated from GWAS variants. These results suggested that ABC variants could potentially aid in precision prevention and early detection of CRC.

ORs of CRC risk for each PRS decile in the samples from GECCO cohort. PRSice software was used to calculate the ORs adjusted for sex and age. All the tests were two-sided. The solid dots in the center for the error bars indicated the OR values, and the error bars were the corresponding 95% confidence intervals of the ORs.

2. The authors compared the ABC model with other functional annotations (closest gene, eQTL gene, PC-HiC, etc) in the accuracy for assigning genes to regulatory variants (Figure 3e) and concluded that the ABC model had better recall accuracy. It is interesting for the authors to compare the ABC model with the reported functional annotation models, such as INQUIST (Nature 2017. 551(7678):92-94), IMPACT (Nat Genet 2020. 52(12):1346-1354) and PAINTOR with PageRank (Nat Genet 2021. 53(3):392-402).

Response: Thanks for the meaningful comment. We further compared the ABC model with other functional annotation models, including INQUIST, IMPACT and PAINTOR.

INQUISIT, which integrates various genomic information sources such as chromatin interactions, enhancer-promoter correlations, eQTLs, transcription factor binding, gene expression, and topologically associated domain boundaries, is primarily designed to rank target genes at GWAS loci. However, it couldn't provide predictions for functional variants. On the other hand, the ABC model focuses on predicting enhancer-gene connections, enabling the identification of risk variants and the prediction of target genes.

IMPACT is a method that predicts cell-state-specific regulatory elements based on epigenomic and sequence profiles of experimental cell-state-specific TF binding. However, it does not consider chromatin interaction and is unable to predict the target genes of regulatory variants.

PAINTOR is a fine-mapping strategy that integrates external functional annotations with genetic association data to improve the prioritization of causal variants. To compare the performance of PAINTOR and the ABC model in predicting functional variants, we obtained the credible variants predicted by PAINTOR from CAUSALdb (<http://www.mulinlab.org/causaldb/index.html>). We then evaluated the recall rate of functional variants within a credible set of 27 variant-gene connections, which were previously validated by functional experiments in CRC studies. While the ABC model identified 19 variant-gene connections out of 19 cases (70% recall), the PAINTOR only identified 3 functional variants (10% recall). Moreover, PAINTOR is unable to provide predictions of target genes for the functional variants. This analysis demonstrated that the ABC regulatory maps can accurately connect fine-mapped variants to target genes and significantly outperform functional annotation approaches in predicting regulatory connections.

3. The authors have verified ABC variant rs4810856 acts as the causal variant in CRC tumorigenesis by two-stage population study and multipronged experiments. The authors could try to provide additional evidence to support the significant association by other independent populations?

Response: As per the comment. We have validated the association between rs4810856 and CRC risk in GECCO cohort, Chinese population and UK biobank cohort, where the P value was 4.02×10^{-3} , 4.02×10^{-5} and 7.20×10^{-3} . Meanwhile, we explored the association in BioBank Japan Project (BBJ), and the results also illustrated that rs4810856-C allele conferred a significant genetic predisposition to CRC compared with T allele (OR = 1.08, 95%CI = 1.04-1.11, $P = 1.18 \times 10^{-4}$). We added the results in **Supplementary Table 5**.

Result section:

In page 9, line 262, we now state:

“To validate the association between rs4810856 and CRC risk, we tested the risk effect of rs4810856 in multiple CRC European population cohorts, including the meta data (OR=1.03, $P=3.02\times 10^{-2}$), BioBank Japan Project (BBJ) (OR=1.08, $P=1.18\times 10^{-4}$), UK Biobank (OR=1.06, $P=1.02\times 10^{-2}$) and GECCO (OR=1.04, $P=5.59\times 10^{-3}$) (**Supplementary Table 5 and 6**).”

Table S5: Association analyses between rs4810856 and CRC risk in the public datasets

Dataset	Sample (Case/Control)	SNP	MAF	RA	OR (95%CI)	P
Meta (CORECT+CFR1+ MECC1+GECCO)	18,299/19,656	rs4810856	0.32	T	1.03 (1.02-1.04)	3.02×10^{-2}
BBJ	7,062/195,745	rs4810856	0.37	T	1.08 (1.04-1.11)	1.18×10^{-4}
UK Biobank	5,246/31,476	rs4810856	0.31	T	1.06 (1.01-1.10)	1.02×10^{-2}
GECCO	17,789/19,951	rs4810856	0.33	T	1.04 (1.01-1.07)	5.59×10^{-3}

Abbreviations: CORECT, Colorectal Cancer Transdisciplinary; CFR, Colon Cancer Family Registry; MECC, Molecular Epidemiology of Colorectal Cancer; GECCO, Genetics and Epidemiology of Colorectal Cancer Consortium; BBJ, BioBank Japan Project; MAF, minor allele frequency; RA, reference allele; OR, odds ratio; CI, confidence interval.

4. Super-shift EMSA assay conducted in figure 4i indicated that ZEB1 might be the candidate TF which preferentially binds to rs4810856-C allele. However, no specific lines were observed in the figure to represent the enrichment for this TF. An additional ChIP assay of ZEB1 might be helpful to support this issue.

Response: We appreciate the suggestion and have taken it into consideration. Although a specific line indicating the enrichment of ZEB1 in the super-shift EMSA assay was not observed, it was possible that the antibody-TF-DNA complex was too heavy to migrate down in the EMSA/Gel-Shift. However, we did observe a dose-dependent reduction in binding signals at the rs4810856-C allele upon increasing the amount of ZEB1 antibody. Additionally, ChIP-qPCR assays of ZEB1 were performed in SW480 and HCT116 cells, and the results revealed that binding peaks of ZEB1 overlapped with the region containing rs4810856. Furthermore, ZEB1 binding to this region was more significant in SW480 cells carrying rs4810856[CC] than in HCT116 cells carrying rs4810856[CT]. These findings suggested that ZEB1 preferentially binds to the risk allele of rs4810856 to mediate the regulatory activity of the enhancer. The results have been added to the Result section of the revised manuscript.

Result section:

In page 10, line 311, we now state:

“Meanwhile, the ChIP-qPCR results also showed that the binding peaks of ZEB1 overlapped the enhancer region containing rs4810856, and the binding of ZEB1 to this region was more statistically significant in SW480 cells carrying rs4810856[CC] than in HCT116 cells carrying rs4810856[CT]

(Fig. 4j and S10d).”

Chromatin enrichment of ZEB1 at the rs4810856 site measured by ChIP-qPCR assays in SW480 and HCT116 cells. Data were presented as the median (minimum to maximum) and normalized to the input from three repeated experiments, each with three replicates. IgG served as negative control. *** $P < 0.0001$ were calculated by a two-sided Student's t -test.

5. The authors have demonstrated the potential regulatory function of rs4810856 by performing dual-luciferase reporter assays. They showed that the difference of luciferase activity between both alleles was significantly attenuated in a dose-dependent manner upon TF knockdown, indicating the involvement of ZEB1 in the precise regulation of gene expression. Furthermore, similar results were observed in the expression of three target genes in the CRISPR/Cas9 editing cell lines. While the study provides strong evidence for the regulatory function of this variant, it would be beneficial to further support these findings by overexpressing ZEB1 and examining the effects on luciferase activity and target gene expression.

Response: Per the request. We conducted additional dual-luciferase reporter assays upon ZEB1 overexpression in SW480 and HCT116 cells. The luciferase activity of rs4810856[C] increased significantly in a dose-dependent manner, while the luciferase activity of rs4810856[T] remained unchanged, indicating ZEB1 might mediate the regulatory activity of the enhancer in an allele-specific manner. We further investigated the regulatory function of ZEB1 by performing qRT-PCR to evaluate the expression levels of *PREX1*, *CSEIL*, and *STAU1* in CRISPR/Cas9 edited cell lines upon ZEB1 overexpression. The results showed a significant increase of the expression differences between the two alleles upon ZEB1 overexpression, providing additional evidence for the regulatory role of ZEB1 on these target genes. We have included these results in the Results section, along with additional related experiments.

Result section:

In page 10, line 316, we now state:

“In contrast, the differences in luciferase activity between both alleles of rs4810856 was enhanced in a dose-dependent manner when we overexpressed ZEB1 at an increasing dose, suggesting that ZEB1 preferentially binds to the risk allele of rs4810856 in an allele-specific manner (Fig. 4l and S10f).”

The effect of ZEB1 overexpression on relative luciferase activity of vectors containing rs4810856 [C] or rs4810856 [T] allele in SW480 and HCT116 cells. Data were shown as the median (minimum to maximum) from three independent experiments, and each had three replicates. *** $P < 0.0001$, ** $P < 0.001$, * $P < 0.01$ were calculated by a two-sided Student's t -test.

In page 11, line 346, we now state:

“Noticeably, the expression differences between both alleles were significantly attenuated upon ZEB1 knockdown (Fig. 5e), while the expression differences were significantly increased upon ZEB1 overexpression (Fig. S13a), indicating that rs4810856 acts as an allele-specific enhancer mediated by ZEB1 to directly regulate the expression of three target genes.”

The effects of ZEB1 overexpression on *PREX1*, *CSE1L* and *STAU1* expression in parental (SW480[CC] and HCT116[CT]) and mutated cells (SW480[CT] and HCT116[TT]). Data were presented as the mean \pm SEM from three independent experiments with three technical replicates. *** $P < 0.0001$ was calculated by a two-sided Student's t -test.

6. The authors have identified over 30,000 enhancer-gene connections in CRC tissues by using ABC model, whereas only one of which was validated by large-scale population and experiments with high confidence. The rest of enhancer-gene maps detected by ABC model are encouraged to test the effect of enhancers on gene expression in a high throughput technology.

Response: Thanks for the valuable suggestion. In this study, we prioritized rs4810856 as the most promising variant with the highest ABC score, and we have performed a series of biological experiments to validate its regulatory mechanism. However, we acknowledged that the ABC model used in this study relies on bioinformatics-based predictions, which lack direct experimental

evidence to support enhancer-gene connections. We plan to use the CRISPR-Cas9 genome editing system to test the effect of enhancers on gene expression in future study. We also addressed this limitation in the Discussion section of revised manuscript.

Discussion section:

In page 16, line 534, we now state:

“In addition, the ABC approach was based on computational predictions to identify enhancer-gene connections, and the functional relevance of these connections remains to be experimentally validated. CRISPR-Cas9 genome editing system could be further employed to assess the effect of enhancers on gene expression and support the regulatory connections identified by ABC model.”

7. A body of comprehensive results in the manuscript indicated that variants located in ABC enhancers were enriched in the regulatory region such as transcription factor binding sites and histone modification for active markers. The authors should provide the information of the TF and histone modification that they included in the analyses in detail.

Response: Thank you for the reminding. We obtained ChIP-seq data for TFs and histone modification markers across various cancer cell lines from the ENCODE project. To focus on the relevant TFs, we filtered the dataset to include only those with at least 50,000 peaks in the ChIP-seq data. We have added the information of the TF and histone modification that selected for analysis in the revised manuscript. Additionally, we have included the types of cell lines and tissues used for enrichment analysis in the **Supplementary Table 8**.

Result section:

In page 6, line 150, we now state:

“The ChIP-seq data of histone modification and TFs across various cancer cells were downloaded from ENCODE and TFs with at least 50,000 peaks in the ChIP-seq data were finally selected.”

Minor comments

1. Line 139 “we generated the control variants set (non-ABC variants) with the distribution of MAF and location matched to the set of ABC variants per cancer type”. The authors should state the criterion to generate the non-ABC variants detailly.

Response: We have described the generation of non-ABC variants in detail in the revised manuscript.

Result section:

In page 5, line 143, we now state:

“To characterize features of variants resided in the ABC enhancers (ABC variants), we generated the control variants set (non-ABC variants) using a web tool vSampler with the allele frequencies, number of variants in LD, as well as genomic distribution matched to ABC variants for each cancer type.”

2. The authors conducted quality control for genotype data from GECCO according to following criteria: (1) imputation quality < 0.4; (2) minor allele frequency (MAF) < 1%; (3) deviating from the Hardy-Weinberg equilibrium ($P < 1 \times 10^{-6}$); (4) missing call frequencies > 0.02. (5) mapping to

locations on sex chromosome. Please explain why chose this threshold as the criterion of quality control or add a reference.

Response: Thanks for the comment. The specific quality control criteria for genotype data from GECCO were outlined in an article published in the Journal of the National Cancer Institute (Stephanie et al, *J Natl Cancer Inst*, 2019 Feb 1; 111(2):146-157). We have cited this reference in the revised manuscript.

3. Please label the MAF of rs4810856 C/T allele in the table footnotes in Table 1.

Response: Per the request. The allele frequency of rs4810856-T was 0.35 and we have provided the MAF in **Table 1** in revised manuscript.

4. In the legend of Figure S2 “The circle color indicates the Pearson correlation, and circle size indicates FDR”. FDR abbreviation is not defined. tomorrow

Response: Thanks for the reminder. We have added the full name of FDR in revised manuscript.

5. Although the writing is fluent, many sentences in the manuscript are quite long, the authors are suggested shorten their sentences to improve the readability of the text.

Response: Thanks for the suggest. We carefully shorten the sentences and made the necessary modifications to improve the clarity and accuracy of our expressions. Additionally, we enlisted the assistance of a native speaker, also a professional editor to polish the manuscript.

Reviewer #3 (Remarks to the Author)

In this study, the authors created enhancer-gene maps across 20 cancer types through integrating multi-omics data applying ABC model, providing an informative enhancer-gene connection resource and a novel view of the regulation mechanisms linking non-coding variants that control gene expression to CRC risk. Besides, the variants positioned within ABC enhancers played key roles in tumorigenesis as cancer drivers. Furthermore, they found that target genes affected by ABC variants are closely involved in multiple cancer pathways, high mutation burden, immune cell infiltration and pharmaceutical targets, which will be helpful for suggestive of potential use in clinical practices. They demonstrated an ABC variant rs4810856-C allele contributes to an increased risk of CRC in large-scale multiracial population cohorts. Mechanically, rs4810856 acted as an allele-specific enhancer to distally facilitate expression of PREX1, CSE1L and STAU1, which synergistically activated p-AKT signaling to drive CRC tumorigenesis. This study provides a valuable approach to elucidate the contribution of genetic variants to various traits and diseases, including cancers. The paper is well performed and the data and experimental results shown supports the author's conclusions. I have the following comments.

Response: We are delighted to receive your positive review of our work. We have made corrections to our previous draft according to your suggestions.

Major issues

1. A body of evidence indicated that variants located in ABC enhancers were enriched in the regulatory region within open chromatin status. For example, ABC variants recognized in CRC were remarkably enriched among TF binding sites and active epigenetic makers such as H3K27ac, H3K4me1, ATAC peaks and DNaseI hypersensitive sites (DHSs). I am wondering whether the authors could try to analyze the enrichment of ABC variants among the chromatin state?

Response: Thanks for this suggest. We downloaded core 15 chromatin state from Roadmap and analyzed the enrichment of ABC variants among the chromatin state. The results indicated ABC variants were enriched in active chromatin state, including enhancers and active TSS. We have included these results in the Results section in the revised manuscript.

Result section:

In page 6, line 158, we now state:

“Meanwhile, we analyzed the enrichment of ABC variants among the chromatin state, and found ABC variants were enriched in active chromatin state, including enhancers and active TSS (**Fig. S1**).”

Enrichment analyses of ABC variants in each functional category of chromatin state compared with control variants (non-ABC variants). P values were calculated by two-tailed Fisher's exact test.

2. The results presented that ABC variants were significant enriched in CRC GWAS loci and ABC genes were associated with the clinical utility including multiple cancer pathways, high mutation burden, immune cell infiltration and pharmaceutical target, highlighting the importance of the connections predicted by ABC model. Accordingly, I am interested to see whether ABC variants might serve as an effective tool for early screening of CRC?

Response: Thanks for the comment. To evaluate the potential of ABC variants for early screening of CRC, we conducted polygenic risk score analyses using PRSice in the GECCO cohort. Our findings showed that individuals in the top 20% of the PRS calculated from ABC variants had a 1.36-fold higher CRC risk (95% CI = 1.27-1.45) compared to those in the bottom 20%, which was a stronger association compared to the PRS calculated from GWAS variants. These results suggest that ABC variants hold promise for precision prevention and early detection of CRC.

ORs of CRC risk for each PRS decile in the samples from GECCO cohort. PRSice software was used to calculate the ORs adjusted for sex and age. All the tests were two-sided. The solid dots in the center for the error bars indicated the OR values, and the error bars were the corresponding 95% confidence intervals of the ORs.

3. The authors have conducted the heritability analyses of ABC variants across 20 cell types, including CRC cells, “ABC variants could explain a significant fraction of cancer heritability, ranging from 0.5% in thyroid carcinoma (THCA) to 12% in prostate adenocarcinoma”. Authors can analyze the heritability of ABC variants derived from CRC tissues.

Response: According to your suggestion, we used LD score regression to evaluate the contribution of ABC variants to the heritability of CRC in our tissue samples. The result showed that ABC variants could explain a significant fraction (1.66%) of cancer heritability, indicating their potential role in the development of CRC.

Result section:

In page 8, line 246, we now state:

“Moreover, we performed in-depth analysis using LDSC to assess the proportion of ABC variants associated with heritability for CRC, and found that ABC variants could explain a significant fraction (1.66%) of cancer heritability (**Fig. S8b**).”

Proportion of GWAS trait heritability of CRC explained by ABC variants. The error bars represented standard error.

4. The authors performed the dual-luciferase reporter assays and EMSA assay to verify that ZEB1 might be the candidate TF which preferentially bounds to the risk allele of rs4810856. To furthermore assure that ZEB1 is the key regulatory factor of the process that rs4810856 influences the expression of three oncogenes, authors are encouraged to perform a ChIP assay in both CRC cell lines used in the EMSA assay.

Response: Thank you for the meaningful suggest. We conducted ChIP-qPCR assays of ZEB1 in SW480 and HCT116 cells to evaluate the regulatory activity of the rs4810856. The results revealed that the binding peaks of ZEB1 overlapped with the region containing rs4810856. Notably, the binding of ZEB1 to this region was more significant in SW480 cells carrying rs4810856[CC] than in HCT116 cells carrying rs4810856[CT], indicating rs4810856 might act as an allele-specific enhancer which was mediated by ZEB1. We have added these findings in the Results section of the revised manuscript.

Result section:

In page 10, line 311, we now state:

“Meanwhile, the ChIP-qPCR results also showed that the binding peaks of ZEB1 overlapped the enhancer region containing rs4810856, and the binding of ZEB1 to this region was more statistically significant in SW480 cells carrying rs4810856[CC] than in HCT116 cells carrying rs4810856[CT] (Fig. 4j and S10d).”

Chromatin enrichment of ZEB1 at the rs4810856 site measured by ChIP-qPCR assays in SW480 and HCT116 cells. Data were presented as the median (minimum to maximum) and normalized to the input from three repeated experiments, each with three replicates. IgG served as negative control. *** $P < 0.0001$ were calculated by a two-sided Student's t -test.

5. The results showed in single variant-CRISPR/Cas9 editing cell lines are very interesting and adequately support the direct effects of the ABC variant rs4810856 on expression of three target genes and cell proliferation, which are mediated by ZEB1. The remarkable effects of TF ZEB1 on the expressions of three target gene were observed under the consideration of ZEB1 knockdown, the effects of ZEB1 overexpression on it are warranted in the single variant-CRISPR/Cas9 editing cell lines.

Response: Thank you for the meaningful suggest. We performed qRT-PCR assays to assess the expression level of *PREX1*, *CSE1L* and *STAU1* in the single variant-CRISPR/Cas9 editing cell lines upon *ZEB1* overexpression. The results demonstrated a significant increase of expression differences between the two alleles upon *ZEB1* overexpression. We have added the relevant experiments in the Results section of the revised manuscript.

Result section:

In page 11, line 346, we now state:

“Noticeably, the expression differences between both alleles were significantly attenuated upon ZEB1 knockdown (Fig. 5e), while the expression differences were significantly increased upon ZEB1 overexpression (Fig. S13a), indicating that rs4810856 acts as an allele-specific enhancer mediated by ZEB1 to directly regulate the expression of three target genes.”

The effects of ZEB1 overexpression on *PREX1*, *CSE1L* and *STAU1* expression in parental (SW480[CC] and HCT116[CT]) and mutated cells (SW480[CT] and HCT116[TT]). Data were presented as the mean \pm SEM from three independent experiments with three technical replicates. *** $P < 0.0001$ was calculated by a two-sided Student's *t*-test.

6. Furthermore, to test the direct effects of the ABC variant rs4810856 on cell proliferation whether are dependent on the ZEB1, the effects of ZEB1 knockdown or overexpression on cell proliferation by CCK8 and colony formation assays are encouraged to conduct in the single variant-CRISPR/Cas9 editing cell lines with different rs4810856 genotypes.

Response: As recommended, we performed CCK-8 assays in the single variant-CRISPR/Cas9 editing cell lines upon ZEB1 overexpression or knockdown. The results demonstrated that the overexpression of ZEB1 significantly promoted the proliferation of the parental cells compared to the mutated cells, while the knockdown of ZEB1 markedly suppressed the proliferation of the parental cells compared to the mutated cells, suggesting that the risk C allele of rs4810856 might function as an allele-specific enhancer promoting CRC cell proliferation via ZEB1 mediation. We have included these findings in the Results section of the revised manuscript.

Result section:

In page 11, line 361, we now state:

“Meanwhile, we observed a significant attenuation of the differences in cell proliferation between both alleles upon ZEB1 knockdown, whereas these differences were significantly increased upon ZEB1 overexpression (Fig. S13b and S13c).”

The direct effect of rs4810856 genotype on cell proliferation upon ZEB1 knockdown or overexpression. Results were shown as the means \pm SEM from triplicate experiments. *** $P < 0.0001$, * $P < 0.01$ were calculated by a two-sided Student's *t*-test.

7. It is ingenious that the authors found the critical mRNA PTEN which provided the clues for the associations between STAU1 and p-AKT signaling pathway in RIP assay in Fig 6a and Fig 6b. The authors should provide the list of the potential target mRNA that could bind to STAU1 in the supplementary material.

Response: Thank you for the suggest. We performed RIP-seq with anti-STAU1 antibody in SW480 cells and successfully identified 1,341 candidate target mRNAs. The list of the potential target mRNAs was then provided in **Supplementary Table 7**.

8. Since CREs are commonly cell-type-specific with the rapid development of scATAC-seq technology, understanding cancer regulatory mechanisms at the cell type-specific level is required to fully interpret the functional impact of risk variants. Could the authors consider that integrate single-cell sequencing data into the ABC model to identify cell-type specific enhancer-gene connections?

Response: Thank you for the valuable comment. We would like to mention that our study found that only 0.5% of enhancer-gene connections were shared among pairs of cancer types, indicating a highly cancer-type-specific regulatory landscape. While it was desirable to construct enhancer-gene connections at the single-cell level, the limited resources of single-cell omics-data have made it temporarily challenging. However, we are optimistic that with the rapid development of single-cell sequencing technologies, such as scRNA-seq and scATAC-seq, we will be able to construct cell-type-specific regulatory maps in future research. We also addressed this limitation in the Discussion section of revised manuscript.

Discussion section:

In page 16, line 530, we now state:

“First, cis-regulatory elements are commonly cell-type-specific, and to fully understand the molecular mechanisms underlying cancer risk, it was desirable to dissect enhancer-gene connections at the cell-type-specific level using single-cell technology. However, due to the current limitations in resources for single-cell omics-data, it is challenging to construct comprehensive regulatory maps at the single-cell level.”

Minor issues

1. Line 139 “we generated the control variants set (non-ABC variants) with the distribution of MAF and location matched to the set of ABC variants per cancer type”. It was unclear which method the author used to generate the non-ABC variants.

Response: Thanks for the comment. We have described the generation of non-ABC variants in detail in the revised manuscript.

2. Line 796 “Replicability of enhancer-gene pairs across cancer types”. The authors should describe the method that they conducted to this analysis of Fig 1h detailly in the method.

Response: Thanks for the comment. We have added a description of the correlation analysis method to this figure legend in the revised manuscript.

3. Line 878 “Genes PREX1, CSE1L and STAU1 are in the same TAD”. While the statement in the main text is “The results showed that rs4810856 harboring enhancer and three target genes were present in the same TAD”. The authors should refine this description in the figure legend of Fig 4d.

Response: Sorry for the confound. We have clarified the description in the revised manuscript.

In page 28, line 937, we now state:

“Topologically Associated Domain (TAD) overlaid with gene annotations surrounding rs4810856 as seen from the Hi-C data in HCT116 cells. The enhancer containing rs4810856 and three target genes were present in the same TAD.”

4. The authors should add the types of cell lines or tissues that TF and histone markers downloaded from ENCODE database in the supplementary material.

Response: Thanks for the comment. We downloaded the ChIP-seq data of TF and histone modification of all cancer cells from ENCODE, and we finally included the markers with at least 50,000 peaks in each ChIP-seq data. Additionally, we have included the types of cell lines and tissues used for enrichment analysis in the **Supplementary Table 8**.

5. While the manuscript was understandable, the English writing still need some polish.

Response: Thanks for the suggest. We have carefully checked the grammar and sentence errors and modified the manuscript accordingly. Furthermore, we invited a native speaker, also a professional editor to polish our manuscript, and we hope the revised manuscript will be clearer and more accurate on expressions.

Reviewer #4 (Remarks to the Author)

This is an interesting work and Ying et.al systematically established genome-wide enhancer-gene maps across 20 human cancer types by integrating the multi-omics data using the ABC model. Numerous enhancer-gene connections were identified with potential regulatory functions associated cancer development, which provided the important insights into assigning the non-coding variants to underlying target genes. Notably, they illustrated that ABC model performed better in linking regulatory variants to target genes than other previous approaches. The authors further focused on an ABC variant rs4810856 which was significantly associated with CRC risk, and claimed the functional mechanism using a series of biological assays. Interestingly, they reported a regulatory role of this variant on three target genes which could exert synergistic effects to activate p-AKT signalling. The work presented was of high quality and would be of broad interest, providing regulatory maps linking causal variants to cancer and indicating a novel mechanism among cancer growth. The paper is well performed. However, there are some results need further development to support the author's conclusions before publication.

Response: Thanks for these positive comments. We have addressed the concerns in the following responses.

major issues

1. The authors used ABC model systemically integrating several multi-omics data to build genome-wide enhancer-gene maps in 20 cancer types, and found the most regulatory landscapes were highly cancer-type-specific in figure 1h. However, it needs to discuss in-depth about the probable reasons of this tumor specific issue.

Response: Thank you for the meaningful comment. In this study, we observed that only 0.5% of enhancer-gene connections were shared among pairs of cancer types, indicating a highly cancer-type-specific regulatory landscape. This can be attributed to the fact that cis-regulatory elements are often cell-type-specific, meaning they are only active in certain cell types and not others. For example, Lijin et al. (*Nat Commun.* 2021 Mar 3;12(1):1419) used an integrative analysis of histone modification profiling and epigenetic and genomic features to gain insight into dynamic regulation of active and inactivate chromatin states across a diverse set of 60 cancer cell lines, and found high variability in histone modifications that were associated with enhancer or promoter regions. Additionally, different cancer types can have distinct genetic alterations and signaling pathways, further contributing to the cancer-type-specific regulatory landscape. We have elaborated on these findings in the Discussion section of our manuscript.

Discussion section:

In page 13, line 427 we now state:

“Notably, we found only 0.5% of enhancer-gene connections were shared among pairs of cancer types, indicating that most regulatory landscapes were highly cancer-type-specific, which is consistent with the findings reported by Lijin et al (*Nat Commun.* 2021 Mar 3;12(1):1419). This cancer-type specificity can be attributed to the fact that cis-regulatory elements are often cell-type-specific and exhibit activity only in certain cell types. Furthermore, the distinct genetic alterations

and signaling pathways in different cancer types also contribute to the cancer-type-specific regulatory landscape (*Nature*. 2015 Feb 19; 518(7539):317-30).”

2. The authors used an average Hi-C data in some cancer types that lacking cell-type specific Hi-C to build enhancer-gene maps. This might raise concern about whether this average Hi-C data were suitable to compute cancer-specific enhancer-gene maps. The authors should explain the rationality of this replacement.

Response: Thank you for the comment. We used averaged Hi-C contacts to supplement our analysis to account for the lack of Hi-C data in the specific cancer type. The average Hi-C matrix as the arithmetic mean of the 10 cell-type-specific Hi-C matrices was computed by Joseph et al (*Nature*. 2021 May;593(7858):238-243). They found that the ABC model can accurately predict cell-type-specific enhancer-gene connections using this averaged Hi-C dataset, which performed similarly to using cell-type-specific promoter capture Hi-C data. Therefore, the ABC model can be used to predict enhancer-gene connections in a given cell type even in the absence of cell-type-specific Hi-C data by using averaged Hi-C data. We have explained the rationality of using average Hi-C data to generate enhancer-gene maps across various cancers in the revised manuscript.

Method section:

In page 17, line 575 we now state:

“The average Hi-C maps at 5 kb resolution were generated from 10 human cell types and were found to show a strong correlation with cell-type-specific Hi-C maps.”

3. The target genes affected by ABC SNPs were found to related to cancer signaling pathways, high mutation burden, frequent amplification and deletion alterations, immune infiltration and pharmaceutical targets, indicating the target genes were quite important in cancer development. It might be more meaningful to discuss the potential clinical application of these target genes in the discussion.

Response: Thank you for the valuable suggestion. We have added more descriptions to discuss the potential clinical application of these target genes affected by ABC enhancers in the Discussion section.

Discussion section:

In page 13, line 447 we now state:

“Genetic risk factors can contribute to the development of cancers, some of which were potentially modifiable (*Nature*. 2016 Jan 7;529(7584):43-7). Therefore, it is crucial to understand the molecular targets of known carcinogens in order to design effective therapeutic interventions. we found that the target genes affected by ABC enhancers were closely related to cancer signaling pathways, high mutation burden, immune infiltration, and pharmaceutical targets. These results indicated the importance of the ABC genes which might have significant clinical applications. Numerous studies have demonstrated the critical role of the immune system in cancer development and progression. For instance, immune infiltrates have been identified as an integral component of the tumor

microenvironment, and have been shown to play a critical role in tumor progression, responses to immunotherapy and clinical outcomes (*Nat Rev Cancer*. 2012 Mar 15;12(4):298-306). In our study, the immune infiltration-associated genes identified by the ABC approach may provide promising therapeutic strategies for cancers. Furthermore, our discovery of target genes involved in cancer signaling pathways, high mutation frequency, and pharmaceutical targets might enable more precise therapeutic targeting of cancers.”

4. The authors performed a super-shift EMSA assay to confirm that ZEB1 is the candidate TF that preferentially binds to the C allele of rs4810856. The results, as shown in Figure 4i, indicate that the binding band at the C allele was weakened in a dose-response manner with increasing amounts of ZEB1 antibody. However, the figure does not appear to show any super-shift lines. The authors should provide an explanation for this.

Response: We thank you very much for this comment. Although we did not observe a specific line indicating the enrichment of ZEB1 in the super-shift EMSA assay, which might possible that the antibody-TF-DNA complex was too heavy to migrate down in the EMSA/Gel-Shift. Nevertheless, we did observe a dose-dependent reduction in binding signals at the rs4810856-C allele upon increasing the amount of ZEB1 antibody. Meanwhile, we performed ChIP-qPCR assay of ZEB1 in CRC SW480 and HCT116 cells, and the results showed that binding peaks of ZEB1 overlapped the region containing rs4810856. Consistently, the binding of ZEB1 to these this region was more significant in SW480 cells carrying rs4810856[CC] than in HCT116 cells carrying rs4810856[CT]. These findings suggested that ZEB1 preferentially bound to the risk allele of rs4810856. We also added these results to the Result section in revised manuscript.

Result section:

In page 10, line 311, we now state:

“Meanwhile, the ChIP-qPCR results also showed that the binding peaks of ZEB1 overlapped the enhancer region containing rs4810856, and the binding of ZEB1 to this region was more statistically significant in SW480 cells carrying rs4810856[CC] than in HCT116 cells carrying rs4810856[CT] (Fig. 4j and S10d).”

Chromatin enrichment of ZEB1 at the rs4810856 site measured by ChIP-qPCR assays in SW480 and HCT116 cells. Data were presented as the median (minimum to maximum) and normalized to the input from three repeated experiments, each with three replicates. IgG served as negative control. *** $P < 0.001$

0.0001 were calculated by a two-sided Student's *t*-test.

5. The authors conducted dual-luciferase reporter assays to claim the potential regulatory function of the variant. Luciferase activity between both alleles of this variant was significantly attenuated in a dose-dependent manner upon TF knockdown, which provided important evidence to support that ZEB1 might involve in the precise regulation of gene expression. Besides, the effects of luciferase activity in this enhancer region based on overexpression of the TF still needed to be assessed.

Response: To address the comment, we conducted additional dual-luciferase reporter assays upon overexpression of the ZEB1 to evaluate the effects of regulatory activity in the enhancer region containing rs4810856. The results showed that luciferase activity of rs4810856[C] was significantly increased in a dose-dependent manner upon overexpression of the TF, indicating that ZEB1 may mediate the regulatory activity of the enhancer containing rs4810856. We further added related experiments to the Result section in revised manuscript.

Result section:

In page 10, line 316, we now state:

"In contrast, the differences in luciferase activity between both alleles of rs4810856 was enhanced in a dose-dependent manner when we overexpressed ZEB1 at an increasing dose, suggesting that ZEB1 preferentially binds to the risk allele of rs4810856 in an allele-specific manner (Fig. 4l and S10f)."

The effect of ZEB1 overexpression on relative luciferase activity of vectors containing rs4810856 [C] or rs4810856 [T] allele in SW480 and HCT116 cells. Data were shown as the median (minimum to maximum) from three independent experiments, and each had three replicates. *** $P < 0.0001$, ** $P < 0.001$, * $P < 0.01$ were calculated by a two-sided Student's *t*-test.

6. The ABC SNPs were indicated to be enriched in GWAS region, and could explain a considerable proportion of cancer heritability in figure 2d and 2e. This was an interesting finding that links ABC SNPs to cancer risk and represents these SNPs identified in this study might important role in cancer progression. However, it should describe better how these GWAS variants were selected for each cancer types in the method.

Response: In our study, the GWAS summary statistics for each cancer type were downloaded from

the GWAS catalog and we included SNPs reached genome-wide significant ($P < 5 \times 10^{-6}$) for enrichment analyses. Meanwhile, GWAS loci was defined as the genomic region containing SNPs in LD with the index SNP at $r^2 \geq 0.2$. We have described the selection of GWAS variants in the revised manuscript.

Supplementary Methods section

“We obtained the GWAS summary statistics for each cancer type from the GWAS catalog and selected SNPs that achieved genome-wide significance ($P < 5 \times 10^{-6}$) for subsequent enrichment analyses. GWAS loci was then defined as the genomic region encompassing SNPs in linkage disequilibrium (LD) with the index SNP at $r^2 \geq 0.2$.”

7. The authors performed association analysis and found the significant association between the variant and CRC risk in three independent population including European and Chinese ethnic, which provided strong clues to confirm the genetic susceptibility of rs4810856 to CRC. However, only additive model was used to assess the association in European population. Other genetic models including allelic, dominant and recessive still needs to be conducted in this population.

Response: Thanks for the suggest. We have added allelic, dominant and recessive model to analyse the association between rs4810856 and CRC risk in European population including GECCO and UK biobank cohort. The results were consistent with the additive model. We have added these results in **Supplementary Table 6** in the revised manuscript.

Table S6. Association analyses between ABC variant rs4810856 and CRC risk in the individuals from UK Biobank and GECCO

Variant	UK Biobank (N=36,722)			GECCO (N=37,740)			
	Cases/ Controls	OR (95% CI) [†]	P*	Cases/ Controls	OR (95% CI) [†]	P*	
rs4810856	TT	497/3,337	1.00 (Ref)		1.00 (Ref)		
	CT	2,175/12,528	1.08 (0.97-1.21)	0.141	7,576/8,467	1.08 (1.01-1.16)	3.70×10 ⁻²
	CC	2,572/14,603	1.16 (1.05-1.29)	5.10×10 ⁻³	8,490/9,318	1.11 (1.03-1.19)	3.68×10 ⁻³
	Dominant		1.11 (1.04-1.17)	8.12×10 ⁻⁴		1.04 (1.00-1.09)	3.90×10 ⁻²
	Recessive		1.14 (1.03-1.26)	1.12×10 ⁻²		1.10 (1.02-1.18)	8.22×10 ⁻³
	Additive		1.07 (1.03-1.11)	1.46×10 ⁻³		1.04 (1.01-1.07)	5.59×10 ⁻³

Abbreviations: OR, odds ratio; CI, confidence interval.

[†]ORs and 95% CIs calculation were conducted under assumption that variant alleles were risk alleles.

*All P values were calculated by unconditional logistic regression model after adjusting for gender, age, smoking status and drinking status.

8. Smoking and drinking status are collected and adjustment in the association analysis to exclude the confounding of common risk factors for CRC. However, there were many important environmental factors, including physical activity, dietary factors, the regular use of aspirin or other anti-inflammatory drugs involved in CRC development. It might be meaningful including these confounds in logistic regression to assess more accurate association between the candidate variant and CRC risk.

Response: Thanks for the valuable comment. We fully understand it might be better to include more environmental factors to support the conclusion of association study. However, due to the incomplete demographic and environmental information of the study population, here only gender, age group, smoking and drinking status were including for the adjustment. We will try our best to collect enough environment factors to support our results and also discussed this limitation in the Discussion section of our revised manuscript.

Discussion section:

In page 16, line 541, we now state:

“Finally, more environmental and lifestyle factors such as physical activity, dietary factors, the regular use of aspirin or other anti-inflammatory drugs should be adjusted to further improve the association determination.”

minor issues

1. The authors stated that ABC model performed better in predicting regulatory elements and their target genes compared with other approach, but it still needs to describe more detail about this model in the method.

Response: Thanks for the suggest. We have provided a more detailed description of the ABC model in the Method section.

2. The frequency of the risk allele of rs4810856 in Chinese population should be provided in the results.

Response: As per the comment, we have added the MAF of rs4810856 in Chinese population in **Table 1**.

3. The identity of cell lines used and test free of mycoplasma should be stated in the manuscript.

Response: Thanks for the comment. This statement has been added in the revised manuscript.

4. The western-blot lines showed in Figure 6E were too narrow. There is a need to reserve more space above and below the lines.

Response: We thank you for raising this point. We have adjusted the western-blot panel with wider space above and below the lines in the revised manuscript.

5. While the manuscript was understandable, the English writing still need some polish.

Response: Thanks for the suggest. We carefully checked the grammar and sentence errors in our manuscript and made the necessary modifications. Additionally, we sought the assistance of a native speaker, also a professional editor to further improve the clarity and accuracy of our expressions.

REVIEWER COMMENTS

Reviewer #1 (Remarks to the Author):

This paper is improved but I do not find many of the assertions very convincing. Aside from the narrative remaining somewhat confused as I stated in the original criticism the data supporting rs4810856 as a cancer risk factor does not meet the required statistical significance. Furthermore, the box plots of gene expression (panel c) are not terribly convincing. The authors should attend to various typos that litter the text.

Reviewer #2 (Remarks to the Author):

The authors have well addressed my concerns. I have two minor comments.

1. The PRS derived from ABC variants exhibits enhanced predictive capacity for CRCs compared to the PRS constructed from GWAS variants. Considering that the majority of causal variants are shared across diverse ethnic populations, and PRS derived from these variants may enhance its trans-ancestry portability (Am J Hum Genet 2020;106:805-817; Nat Genet 2020;52:1346-1354), it is worth investigating the performance of the PRS derived from ABC variants in predicting CRCs across different populations.

2. CRC ABC variants contain reliable functional genetic variants, and the ABC model demonstrates better predictive capability for functional variants among the given set of 19 functional variations compared to PAINTOR (16/19 vs. 3/19). However, the performance of IMPACT in identifying these variants remains unknown. In INQUIST, the target genes are predicted by combining risk variants data with multiple sources of genomic information, including chromatin interactions, computational enhancer-promoter correlations, eQTL results, TF binding, gene expression and TAD boundaries (Nature 2017. 551(7678):92-94: 5th paragraph and INQUISIT pipeline section). The score of a target gene is determined based on the number of functional evidences that both the gene and its corresponding variants hit. This enables the identification of target genes along with paired variants contributing to the score. Among the 19 variant-gene connections mentioned by the authors in this study, how does the accuracy of precision-recall in INQUISIT compare to the ABC model?

Reviewer #3 (Remarks to the Author):

All questions have been addressed.

Reviewer #4 (Remarks to the Author):

The authors have invested substantial effort in addressing the concerns that were raised. They have made noteworthy progress in resolving the majority of the issues. However, a few minor matters still require attention and would greatly benefit from further modifications.

1. The authors conducted a thorough quality assessment of the high-throughput sequence data from 10 colorectal cancer (CRC) tissues, examining key metrics such as Q30 base fraction, GC content, and mapped reads fraction. The authors could further analyze the correlation between samples, which would serve as an additional means to evaluate the quality of the sequencing data.

2. The text contains a few spelling mistakes, such as the word "from" being typed as "form" and contains several lengthy sentences that is not benefit for readability. Please check it carefully.

REVIEWER COMMENTS

Reviewer #1 (Remarks to the Author):

This paper is improved but I do not find many of the assertions very convincing. Aside from the narrative remaining somewhat confused as I stated in the original criticism the data supporting rs4810856 as a cancer risk factor does not meet the required statistical significance. Furthermore, the box plots of gene expression (panel c) are not terribly convincing. The authors should attend to various typos that litter the text.

Response: Thanks for your valuable insights of our work. We have taken your comments into serious consideration and have made improvements to address the concerns raised in the revised manuscripts.

Given the major aim of this work was to provide a comprehensive landscape for regulatory maps linking regulatory variants to their potential target genes in human cancers, we placed a particular emphasis on exploring genetic variants with potential regulatory function. Specifically, we focused on rs4810856 that exhibited the highest ABC score, highlighting its considerable potential in CRC. We value your point regarding the statistical significance of the evidence supporting rs4810856 as a cancer risk factor. To address this concern thoroughly, we conducted rigorous validations of the association between the variant rs4810856 and CRC risk in multiple independent cohorts, encompassing European and Chinese populations. Even though the P value did not reach 1×10^{-8} , the comprehensive analyses consistently supported the association of rs4810856 with CRC risk (**Table 1 and Supplementary Table 5 and 6**). Consequently, we have tone down our conclusions as appropriate to suggest that rs4810856 is associated with CRC risk in the revised manuscript.

Furthermore, we conducted eQTL analysis between rs4810856 and the three target genes in our own CRC tissues. The results indicated that the C genotype of rs4810856 was association with higher expression levels of the three genes (**Figure 4c**). Based on the ABC predictions as well as the eQTL results, we further conducted CRISPR/Cas9-mediated genome editing approach to precisely edit the genotype of rs4810856 in CRC cell lines. Subsequent qRT-PCR experiments directly confirmed that the C>T change led to a significant decrease in the expression levels of the three target genes (**Figure 5d**). These findings provided solid and direct evidence supporting the regulatory role of rs4810856 on the expression of these three genes and therefore contributing to CRC risk.

Meanwhile, we have carefully examined the manuscript and rectified any mistake identified.

4c. The gene expression levels were normalized relative to *GAPDH*, and the associations between the SNP genotype and gene expression levels were assessed using a linear regression model. Data are shown as the median (minimum to maximum). All *P* values are calculated by linear regression analysis.

5d. Expression of *PREX1*, *CSE1L* and *STAU1* in parental (SW480[CC] and HCT116[CT]) and mutated cells (SW480[CT] and HCT116[TT]). Data were presented as the mean \pm SEM from three independent experiments with three technical replicates. ****P*<0.0001, ***P*<0.001 were calculated by a two-sided Student's *t*-test.

Reviewer #2 (Remarks to the Author):

The authors have well addressed my concerns. I have two minor comments.

1. The PRS derived from ABC variants exhibits enhanced predictive capacity for CRCs compared to the PRS constructed from GWAS variants. Considering that the majority of causal variants are shared across diverse ethnic populations, and PRS derived from these variants may enhance its trans-ancestry portability (Am J Hum Genet 2020;106:805-817; Nat Genet 2020;52:1346-1354), it is worth investigating the performance of the PRS derived from ABC variants in predicting CRCs across different populations.

Response: Thanks for the meaningful suggestion. We have performed polygenic risk score analyses in PLCO population, consisting of 1,233 cases and 8,631 controls. Our findings revealed that participants in the top 20% of the PRS calculated from ABC variants exhibited a 1.50-fold CRC risk (95% CI= 1.23-1.83) compared with those in the bottom 20%, which was higher than the PRS calculated from GWAS variants. These results were consistent with those in the GECCO population, and indicated that ABC variants hold potential in enhancing precision prevention and early detection strategies for CRC.

ORs of CRC risk for each PRS decile in the samples from GECCO cohort. PRSice software was used to calculate the ORs adjusted for sex and age. All the tests were two-sided. The solid dots in the center for the error bars indicated the OR values, and the error bars were the corresponding 95% confidence intervals of the ORs.

2. CRC ABC variants contain reliable functional genetic variants, and the ABC model demonstrates better predictive capability for functional variants among the given set of 19 functional variations compared to PAINTOR (16/19 vs. 3/19). However, the performance of IMPACT in identifying these variants remains unknown. In INQUIST, the target genes are predicted by combining risk variants data with multiple sources of genomic information, including chromatin interactions, computational enhancer-promoter correlations, eQTL results, TF binding, gene expression and TAD boundaries (Nature 2017. 551(7678):92-94: 5th paragraph and INQUISIT pipeline section). The score of a target gene is determined based on the number of functional evidences that both the gene and its corresponding variants hit. This enables the identification of target genes along with paired variants contributing to the score. Among the 19 variant-gene connections mentioned by the authors in this study, how does the accuracy of precision-recall in INQUISIT compare to the ABC model?

Response: Thanks for the comment. As previously mentioned, INQUISIT was designed to rank target genes at GWAS loci, which may not offer predictions for regulatory variants. To evaluate the accuracy of INQUISIT in assigning target genes, we focused on the 19 variants in the credible set that were successfully identified by ABC model. INQUISIT integrated multiple genomic information sources, such as chromatin interactions, enhancer-promoter correlations, eQTLs, transcription factor binding, gene expression, and topologically associated domain boundaries, to predict potential target genes for each variant. However, only 5 genes matched the variant-gene connections as predicted by INQUISIT, while the ABC model could identify 16 variant-gene connections out of 19 cases. Moreover, INQUISIT considers the gene with the highest score as the risk gene, potentially disregarding the fact that one variant might regulate the expression of multiple target genes, which is prevalent in gene expression regulation. This analysis demonstrated that the ABC model accurately connects fine-mapped variants to target genes and significantly outperforms functional annotation approaches, including INQUISIT, in predicting regulatory connections.

Reviewer #3 (Remarks to the Author):

All questions have been addressed.

Response: Thanks for the positive comments. We are grateful for your effort in reviewing and ensuring the quality of our work.

Reviewer #4 (Remarks to the Author):

The authors have invested substantial effort in addressing the concerns that were raised. They have made noteworthy progress in resolving the majority of the issues. However, a few minor matters still require attention and would greatly benefit from further modifications.

Response: Thanks for the overall positive comments. We have addressed the concerns in the following responses.

1. The authors conducted a thorough quality assessment of the high-throughput sequence data from 10 colorectal cancer (CRC) tissues, examining key metrics such as Q30 base fraction, GC content, and mapped reads fraction. The authors could further analyze the correlation between samples, which would serve as an additional means to evaluate the quality of the sequencing data.

Response: Thanks for the valuable comment. We have conducted additional correlation analysis on the sequencing data from 10 CRC samples. The analysis revealed a high correlation among 10 biosamples, which provided insights into the interrelationships among 10 biosamples and underscore their potential shared biological characteristics. We have included these results in the Result section of the revised manuscript.

Result section:

In page 7, line 202, we now state:

“we performed multi-omics analyses of ATAC-seq, H3K27ac ChIP-seq and RNA-seq with high quality from our 10 CRC tissues (Fig. 3a and Fig. S5).”

The correlation of high-throughput sequence data between 10 CRC biosamples were calculated by Pearson correlation analysis using the deeptools.

2. The text contains a few spelling mistakes, such as the word "from" being typed as "form" and contains several lengthy sentences that is not benefit for readability. Please check it carefully.

Response: Thanks, we have carefully examined the manuscript and diligently rectified any errors identified. Additionally, we have also revised lengthy sentences to ensure clarity and coherence in the overall expression.

REVIEWERS' COMMENTS

Reviewer #2 (Remarks to the Author):

All my concerns have been well addressed.

REVIEWER COMMENTS

Reviewer #2 (Remarks to the Author):

All my concerns have been well addressed.

Response: Thanks for the positive comments. We are grateful for your effort in reviewing and ensuring the quality of our work.